# 🌿 AIR: A Systematic Analysis of Annotations, Instructions, and Response Pairs in Preference Dataset

**Bingxiang He**[1][*], **Wenbin Zhang**[2][*], **Jiaxi Song**[1], **Cheng Qian**[4], **Zixuan Fu**[1], **Bowen Sun**[1]
**Ning Ding**[1], **Haiwen Hong**[5], **Longtao Huang**[5], **Hui Xue**[5], **Ganqu Cui**[3][†], **Wanxiang Che**[2][†]
**Zhiyuan Liu**[1], **Maosong Sun**[1]
[1]Tsinghua University, [2]Harbin Institute of Technology, [3]Shanghai AI Lab
[4]University of Illinois Urbana-Champaign, [5]Alibaba Group
hebx24@mails.tsinghua.edu.cn, cuiganqu@pjlab.org.cn
{wenbinzhang,car}@ir.hit.edu.cn

## Abstract

Preference learning is critical for aligning large language models (LLMs) with human values, yet its success hinges on high-quality datasets comprising three core components: Preference **A**nnotations, **I**nstructions, and **R**esponse Pairs. Current approaches conflate these components, obscuring their individual impacts and hindering systematic optimization. In this work, we propose **AIR**, a component-wise analysis framework that systematically isolates and optimizes each component while evaluating their synergistic effects. Through rigorous experimentation, AIR reveals actionable principles: annotation simplicity (point-wise generative scoring), instruction inference stability (variance-based filtering across LLMs), and response pair quality (moderate margins + high absolute scores). When combined, these principles yield +5.3 average gains over baseline method, even with only 14k high-quality pairs. Our work shifts preference dataset design from ad hoc scaling to component-aware optimization, offering a blueprint for efficient, reproducible alignment.

## 1 Introduction

The alignment of large language models (LLMs) with human preferences has become a cornerstone of modern LLM development, achieved through methods like reinforcement learning from human feedback (RLHF; Christiano et al., 2017; Ziegler et al., 2019) and direct preference optimization (DPO; Rafailov et al., 2023). While these techniques differ in their mechanisms—RLHF relies on reward modeling and policy optimization (e.g., PPO; Schulman et al., 2017), while DPO bypasses explicit reward learning—they share a critical dependency: high-quality preference datasets. Whether aligning models for correctness in mathematical reasoning (Shao et al., 2024) or real-world usability in instruction following (Dubois et al., 2024; Lin et al., 2024), the success in powering state-of-the-art models (Dubey et al., 2024; Achiam et al., 2023) has spurred efforts to scale preference dataset construction (Cui et al., 2023; Wang et al., 2024; Xu et al., 2024), yet the principles governing dataset quality remain poorly understood, creating a critical bottleneck in alignment research.

Preference datasets are built on three core components: **Annotations**, **Instructions**, and **Response Pairs**. The construction of preference dataset begins with defining instructions (input prompts that define tasks), which may be synthetic (Xu et al., 2024) or sourced from real human interactions (Zhao et al., 2024). Next, responses are generated—either from models being aligned (on-policy) (Dubey et al., 2024) or external models (off-policy) (Cui et al., 2023). Finally, annotations are collected via human judgments (Bai et al., 2022), AI-based scoring (Cui et al., 2023), or hybrid methods. Some works use pairwise comparisons

---

[*]Equal contribtuion
[†]Corresponding Author.

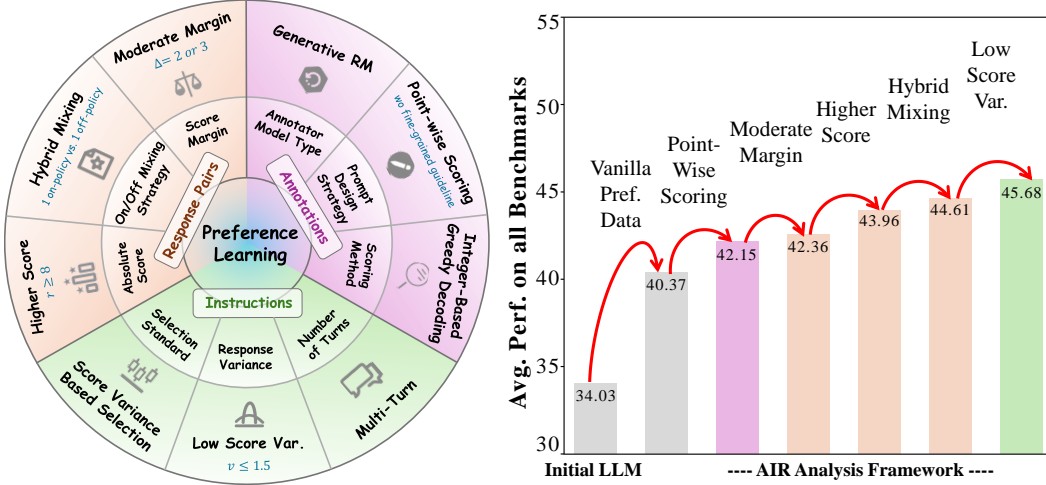

Figure 1: **Left**: The **AIR** framework dissects preference learning into three core components: **A**nnotations, **I**nstructions, and **R**esponse Pairs, with the outermost layer highlighting empirically validated optimal design principles. **Right**: Additive improvement across all benchmarks, resulted by stepwise integrating principles for better annotations, response pairs and instructions with 14k preference pairs.

(Dubey et al., 2024), while others first assign scalar scores via classifier-based (Xu et al., 2024) or generative reward models (RM) (Cui et al., 2023), then construct pairs from score gaps.

However, prior works tend to conflate the impact of these components rather than analyzing them systematically (Yasunaga et al., 2024; Amini et al., 2024). For example, improvements attributed to "better annotations" might instead stem from unmeasured changes in instruction diversity or response quality (Gu et al., 2024). This opacity limits reproducibility and hinders targeted optimization. This conflation obscures a fundamental question: **How do annotations, instructions, and response pairs independently shape dataset efficacy?** Without isolating these components, progress in dataset design remains ad hoc, prioritizing scalability over interpretable principles.

To address this gap, we propose **AIR** (**A**nnotations, **I**nstructions, and **R**esponse Pairs), a component-wise analysis framework to systematically dissect preference dataset construction, as shown in Figure 1 (left). By independently varying each component while controlling others (e.g., fixing instructions and responses to isolate annotation effects), we quantify how design choices in one component propagate to downstream alignment performance. Our goal is to identify generalizable principles for preference dataset design and provide actionable insights for practitioners balancing quality, cost, and scalability.

We conduct controlled experiments to isolate the impact of dataset components using open-source models. Starting with a strong LLM, we generate responses from a diverse pool of LLMs, following a modified UltraFeedback pipeline (Cui et al., 2023). Instructions are sourced from different public datasets, with preferences scored (0–9) via LLMs using multiple annotation strategies. To focus on dataset design, we apply vanilla DPO (Rafailov et al., 2023), avoiding algorithmic confounders. Aligned models are evaluated across 6 commonly used benchmarks spanning instruction following, reasoning, coding, math, and real-world usability, ensuring robust evaluation of alignment-critical capabilities.

Our systematic analysis yields three empirically validated principles:

- **Simplify annotation with generative scoring:** Replace pairwise comparisons or fine-grained guidelines with generative reward models using point-wise scoring—evaluating single responses via greedy decoding. This streamlined approach outperforms classifier-based RMs and complex multi-sample aggregation, prioritizing holistic quality over checklist compliance.

- **Prioritize low-variance instruction selection:** Select instructions with low variance in scores across responses from different LLMs—a proxy for fine-grained preference distinctions—outperforms prior LLM-based methods. Additionally, incorporating multi-turn context offers marginal benefits (MT-Bench Turn 2) but no broad improvement.

- **Optimize response pairs for quality and diversity:** Curate pairs with moderate score margins ($\Delta = 2$ *or* $3$) to balance preference clarity and dataset coverage, high absolute scores ($\geq 8$) to ensure both responses are high-quality, and a hybrid mix of on-policy (base model) and off-policy (external model) responses. This triad maximizes contrastive learning while avoiding noisy or trivial comparisons.

When stepwise integrating our empirically validated principles—generative streamlined annotation (+1.78), variance-based instruction selection (+1.07), and optimized response pairs (+2.46)—we observe a cumulative +5.31 average gain over baseline (w. vanilla preference data) across all benchmarks under controlled variable experiments, as shown in Figure 1 (right). This additive improvement confirms that each AIR component independently contributes to alignment performance while synergizing for compounded gains, validating our framework's systematic design.

## 2 Related Work

**Preference Learning and Alignment.** Post-training alignment via preference learning has become pivotal for refining LLMs. Early work (Christiano et al., 2017; Ziegler et al., 2019) established RLHF as a framework for training reward models and optimizing policies with algorithms like PPO (Schulman et al., 2017). Subsequent innovations simplified the pipeline, notably DPO (Rafailov et al., 2023), which bypasses explicit reward modeling. These methods have driven progress in summarization (Stiennon et al., 2020), instruction following (Ouyang et al., 2022), and frontier LLM development (Team et al., 2023; Achiam et al., 2023; Dubey et al., 2024), but all hinge on the quality of preference datasets—a persistent bottleneck.

**Components of Preference Datasets and Design Insights.** Preference datasets are shaped by three core components: **Annotations, Instructions, and Response Pairs**. Early datasets like Anthropic HH (Bai et al., 2022) relied on human labor for all components, limiting scalability. Recent work automates these elements: UltraFeedback (Cui et al., 2023) generates responses from LLM pools and annotates preferences with GPT-4, while Magpie-DPO (Xu et al., 2024) leverages the chat template to sample on-policy responses and annotates them via classifier-based RMs. Recent efforts have begun dissecting these components. For annotation, Yasunaga et al. (2024) propose probability-based scoring, and Lambert et al. (2024a) observe minimal performance variation across generative LLM-as-a-Judge annotators. For response pair construction, Yu et al. (2025) emphasize long rejected responses, Amini et al. (2024) weight pairs by score margins, while Khaki et al. (2024); Wu et al. (2024) leverage reward distributions. For instruction selection, Kim et al. (2024); Lu et al. (2023) identifies the importance of prompt difficulty filtering, He et al. (2025) highlights the role of data similarity. Broader analyses by Ivison et al. (2024) link dataset quality, algorithms, and reward models to downstream performance. However, these works address components in isolation or conflate dataset design with training algorithms, offering fragmented insights. Our work bridges this gap by proposing a component-wise analysis framework to disentangle and optimize each component, identifying actionable principles for high-quality preference datasets construction.

## 3 Preliminary

### 3.1 Preference Learning

Preference learning aligns language models with human preferences by leveraging pairwise comparisons, most commonly operationalized through the Bradley-Terry (BT) framework (Bradley & Terry, 1952). The BT framework models the probability that response $\mathbf{y}_w$ is preferred over $\mathbf{y}_l$ given instruction $\mathbf{x}$ as:

$$p(\mathbf{y}_w \succ \mathbf{y}_l | \mathbf{x}) = \sigma\left(r^*(\mathbf{x}, \mathbf{y}_w) - r^*(\mathbf{x}, \mathbf{y}_l)\right), \tag{1}$$

where $\sigma$ is the logistic function, and $r^*(\mathbf{x}, \mathbf{y})$ represents the reward score of response $\mathbf{y}$. Modern alignment methods build on this framework in two key paradigms: (1) **RLHF**: Trains a reward model $r_\phi(\mathbf{x}, \mathbf{y})$ to approximate $r^*(\mathbf{x}, \mathbf{y})$, then optimizes policies via RL algorithms like PPO (Schulman et al., 2017); (2) **RL-free**: Bypasses explicit reward modeling by reparameterizing the BT model using the policy $\pi_\theta$ and reference model $\pi_{\text{ref}}$ like DPO (Rafailov et al., 2023):

$$r^*(\mathbf{x}, \mathbf{y}) = \beta \log \frac{\pi_\theta(\mathbf{y}|\mathbf{x})}{\pi_{\text{ref}}(\mathbf{y}|\mathbf{x})}, \tag{2}$$

yielding the DPO objective:

$$\mathcal{L}_{\text{DPO}} = -\mathbb{E}_{(\mathbf{x}, \mathbf{y}_w, \mathbf{y}_l) \sim \mathcal{D}} \log \sigma \left( \beta \log \frac{\pi_\theta(\mathbf{y}_w|\mathbf{x})}{\pi_{\text{ref}}(\mathbf{y}_w|\mathbf{x})} - \beta \log \frac{\pi_\theta(\mathbf{y}_l|\mathbf{x})}{\pi_{\text{ref}}(\mathbf{y}_l|\mathbf{x})} \right). \tag{3}$$

## 3.2 AIR Framework

Both paradigms in Section 3.1 assume access to high-quality preference datasets $\mathcal{D} = \{(\mathbf{x}^{(i)}, \mathbf{y}_w^{(i)}, \mathbf{y}_l^{(i)})\}_{i=1}^M$. However, existing preference learning paradigms treat the dataset as monolithic, conflating three critical components:

- **Annotations**: How to assign scores $\mathbf{s}^{(i)}$ to $(\mathbf{x}^{(i)}, \mathbf{y}^{(i)})$ pairs. We analyze the effect of annotator model types, annotation strategies for prompt design, and scoring methods.
- **Instructions**: How to select high-quality $\mathbf{x}^{(i)}$. We investigate the score variance and number of context turns compared to prior LLM-based methods.
- **Response Pairs**: How to construct $(\mathbf{y}_w, \mathbf{y}_l)$ from scored triples. We study relative score margins, absolute score thresholds, and on/off-policy mixing.

Our **AIR** Framework (**A**nnotations, **I**nstructions, **R**esponse Pairs) systematically isolates each component's impact on alignment performance, shown in Figure 1 (Left), enabling: (1) **Diagnosis**: Identify whether dataset weaknesses stem from noisy annotations, poor instructions, or uninformative responses; (2) **Optimization**: Prioritize improvements to the most impactful component. While validated in LLM alignment, AIR generalizes to any task requiring preference data—from multimodal learning to human-AI collaboration.

# 4 Experiments and Results

We systematically analyze the impact of preference dataset components using our **AIR** framework. Following the experimental setup (Section 4.1), we conduct controlled experiments isolating each component: **A**nnotations (Section 4.2), **I**nstructions (Section 4.3), and **R**esponse Pairs (Section 4.4). We then evaluate their combined impact (Section 4.5) and validate robustness via cross-verification with alternative models and datasets (Section 4.6).

## 4.1 Experimental Setup

We conduct controlled experiments using open-source models to isolate the impact of dataset components following a modified UltraFeedback pipeline (Cui et al., 2023).

**Base Model and Response Generation.** We start with Llama-3.1-Tulu-3-8B-SFT (Lambert et al., 2024a), a strong instruction-tuned model, and generate responses using a diverse pool of 17 open-source LLMs (model list refers to Appendix A.2). These include on-policy responses (from the base model) and off-policy responses from models like Llama, Mixtral, and Qwen2 family.

**Instruction Sources and Annotation.** Instructions are sourced from GPT-4-generated ShareGPT and UltraFeedback, detailed in Appendix A.1.1. We annotate response quality (0–9 scores) using Llama-3.1-70B-Instruct (Dubey et al., 2024) or Qwen2.5-72B-Instruct (Qwen et al., 2025), sampling a subset of 8 responses including 4 on-policy responses using high temperature and 4 off-policy responses per instruction to balance cost and diversity.

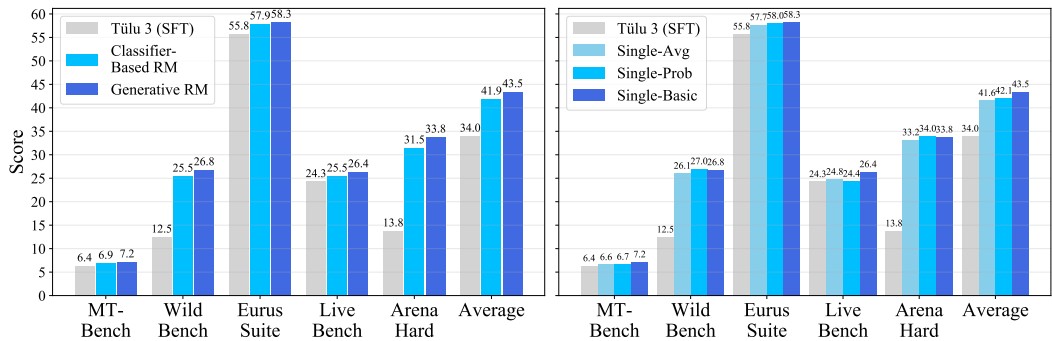

Figure 2: Performance across different benchmarks, comparing different annotator model types (**left**) and different scoring methods (**right**). The average score shows the effectiveness of generative RM and Single-Basic greedy decoding. Scores are normalized to a 100-point scale for averaging.

**Training Protocol.** For all experiments, we sample a total number of 30k preference pairs and apply vanilla DPO (Rafailov et al., 2023) for 1 epoch to isolate dataset effects, avoiding complex RL algorithms. More hyperparameters are in Appendix B.1.

**Evaluation.** We evaluate aligned models across 6 benchmarks: MT-Bench (Zheng et al., 2023b), ArenaHard (Li et al., 2024b), AlpacaEval 2.0 (Dubois et al., 2024), WildBench (V2) (Lin et al., 2024), Eurus evaluation suite (Yuan et al., 2024), and LiveBench (White et al., 2024), covering a various domain among coding, math, reasoning, knowledge, instruction-following, and chat benchmarking. This ensures robustness across alignment-critical capabilities. Benchmark details and evaluation metrics are provided in Appendix B.2.

**Final Dataset Statistics.** We also provide detailed dataset statistics, including distributions of response length and annotation score across LLMs, detailed in Appendix A.4.

## 4.2 Preference Annotation

We analyze three critical dimensions of preference annotation: (1) annotator model types, (2) annotation strategies for prompt design, and (3) scoring methods. Our experiments reveal that:

> 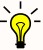 **Takeaway 1**: A **generative RM** with **point-wise scoring** (evaluating single responses directly, avoiding pairwise comparisons or complex guidelines) and **greedy decoding** achieves superior alignment performance, prioritizing simplicity and efficiency over intricate aggregation methods.

### 4.2.1 Annotator Model Type: Classifier-Based RM vs. Generative RM

We compare a state-of-the-art[1] classifier-based RM (Skywork-Reward-Gemma-2-27B-v0.2; RewardBench: 94.3) with a generative RM (Llama-3.1-70B-Instruct; RewardBench: 84.0). Both models score 4 responses per ShareGPT instruction using the point-wise scoring prompt template in Appendix A.3.2. Finally we construct preference pairs for vanilla DPO training.

**Results.** Despite lower RewardBench score, Llama-3.1-70B-Instruct as a generative RM achieves +1.4 higher alignment performance on average across benchmarks, as shown in Figure 2 (left), outperforming the SOTA classifier-based RM on all benchmarks. This suggests that RewardBench may undervalue generative RMs' generalization, and next token prediction (generative RM) better captures holistic response quality.

### 4.2.2 Annotation Strategies: Prompt Design for Response Scoring

Compared to classifier-based RM, generative RM is flexible when annotating the scores, as it can dynamically adjust its annotation prompt. Here we evaluate 6 annotation strategies for

---

[1]Which is SOTA on RewardBench as of our experimental timeframe.

| Annotation Strategy | MT-Bench | WildBench | Eurus | LiveBench | ArenaHard | Avg. |
|---|---|---|---|---|---|---|
| *Tülu-3-8B-SFT (Baseline)* | 6.38 | 12.50 | 55.77 | 24.3 | 13.8 | 34.03 |
| Single-Basic | **7.20** | **26.80** | **58.27** | **26.4** | **33.8** | **43.45** |
| Pair-Basic | 6.73 | 25.49 | 57.86 | 24.8 | 31.9 | 41.47 |
| Pair-Guided | 6.79 | 25.68 | 57.61 | 23.5 | 31.8 | 41.30 |
| Pair-Explained | 6.88 | 25.55 | 57.90 | 23.8 | 32.7 | 41.75 |
| Pair-Guided-Explained | 6.78 | 25.90 | 57.64 | 22.9 | 31.9 | 41.23 |
| Pair-Guided-Explained-FG | 6.60 | 24.56 | 57.68 | 23.8 | 29.6 | 40.33 |

Table 1: Impact of 6 annotation strategies on DPO performance. Blue indicates improvements over the baseline, and orange degradations; color depth reflects the approximate relative magnitude of performance change. All strategies ensure identical instruction sets and pair counts used for DPO training. Scores are normalized to a 100-point scale for averaging. FG denotes Fine-Grained.

generative RM, varying the response count (point-wise vs. pairwise scoring), guideline complexity (none vs. coarse vs. fine-grained), and explanation requirements (presence/absence of reasoning). Details about each strategy and prompt template can refer to Appendix A.3.2.

**Setup.** Using Llama-3.1-70B-Instruct, we score 4 responses per ShareGPT instruction across all strategies, mitigating position bias via reversed-order averaging (Zheng et al., 2023a). For fine-grained guideline setting, task-specific guidelines are generated by LLMs. Annotation strategies range from minimal (Single-Basic: score one response) to complex (Pair-Guided-Explained-Fine-Grained: pairwise scoring with fine-grained guidelines and explanations). Finally we construct preference pairs for vanilla DPO training.

**Results.** The Single-Basic annotation strategy outperforms all alternatives, achieving a top average score of 43.45 (Table 1), surpassing Pair-Basic and the most complex fine-grained guideline approaches by +1.98 and +3.12, respectively. Key insights:

- Pairwise comparisons lag behind, likely due to context overload from dual-response evaluation.

- Fine-grained guidelines harm performance ($\downarrow$0.9), indicating the artifacts in the checklist limit the judge range.

- Explanations/coarse guidelines offer negligible gains ($\leq$ 0.28) over simpler baselines.

These results suggest that minimalist prompt, leveraging annotators' innate judgment, better align with real-world preference signals than complex protocols. The trend toward intricate annotation workflows may inadvertently introduce noise, advocating for streamlined designs.

### 4.2.3 Scoring Methods: Aggregation Techniques for Reliable Annotations

While recent work proposes aggregation methods like multi-sample averaging (Yu et al., 2025) or probability-based weighting (Yasunaga et al., 2024) to improve annotation reliability, we find simple greedy decoding suffices, which is the most robust and well-performed method across all benchmarks.

**Setup.** We compare three scoring methods using point-wise scoring: (1) **Single-Basic**: Greedy decoding (one run, temperature=0); (2) **Single-Avg**: Average scores across 5 runs (temperature=1.0). (3) **Single-Prob**: Probability-weighted scores across token "0" to "9". Detail calculation can refer to Appendix A.3.3.

**Results.** **Single-Basic** achieves the best alignment performance, as shown in Figure 2 (right). While being the simplest, it outperforms Single-Avg by +1.9 and Single-Prob by +1.4 on average. This may be because greedy decoding's robustness stems from its consistency—it avoids noise introduced by stochastic sampling (Single-Avg) or tokenization artifacts (Single-Prob), reflecting the model's deterministic quality assessment.

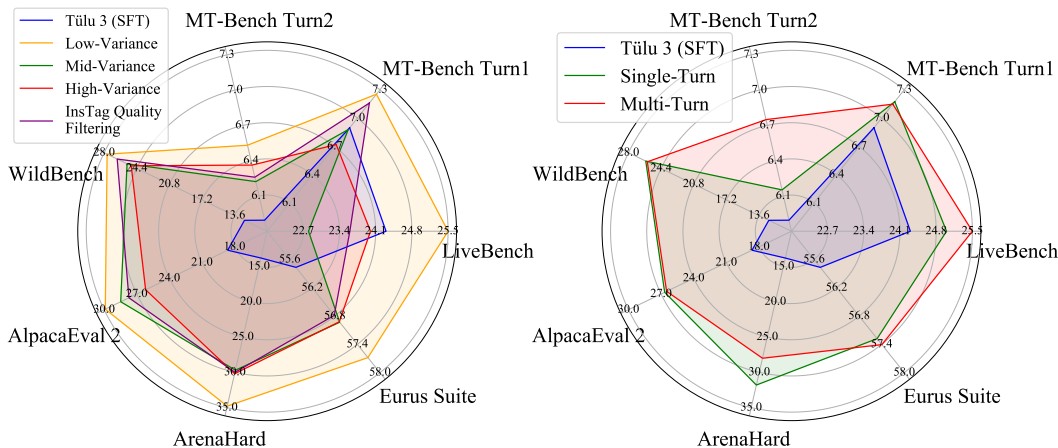

Figure 3: Performance across different benchmarks, comparing different response score variances with top quality tagged by InsTag method (**left**) and number of context turns (**right**). We prioritize instructions with low score variance, while multi-turn context offers marginal gains only on MT-Bench Turn 2 but no broad improvement.

## 4.3 Instruction Selection

Instruction quality is critical for preference learning, yet existing methods prioritize human-curated artifacts (Zhou et al., 2023; Lu et al., 2023), neglecting the preference signal. To better leverage it, inspired by (Li et al., 2024a), we hypothesize that inference-stable instructions which elicit fine-grained distinctions (i.e., small, consistent differences in response quality across LLMs) provide richer signals for alignment. To test this, we propose variance-based instruction selection, prioritizing instructions with low score variance across diverse LLMs. Our experiments demonstrate that:

> 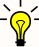 **Takeaway 2**: **Low-variance instructions** outperform high-variance ones across all benchmarks, with variance-based filtering surpassing prior LLM-based methods; multi-turn context provides only marginal MT-Bench Turn 2 gains, prioritizing instruction quality over conversational complexity.

### 4.3.1 Variance-Based Instruction Selection: Fine-Grained Distinctions Matter

To prioritize instructions that expose nuanced distinctions between responses, we propose filtering by score variance. For each instruction $x_i$, we sample $N$ responses from diverse LLMs, annotate their scores $\{s_j\}_{j=1}^N$, and compute the variance $v_i$. Low-variance instructions are prioritized as they indicate fine-grained distinctions critical for alignment.

**Setup.** We sample $N = 5$ responses per instruction, annotate scores via Single-Basic strategy, and calculate variances, detailed distribution can refer to Appendix A.4. We compare three settings: (1) Low: $v_i \leq 1.5$; (2) Mid: $1.5 < v_i \leq 3$; (3) High: $v_i > 3$.

**Results.** Low-variance instructions setting achieves the best performance compared to higher variance ones, as shown in Figure 3 (left), excelling in AlpacaEval 2 (+3.7) and ArenaHard (+4.6). This indicates that low-variance instructions force models to learn subtle, policy-aligned preferences rather than relying on obvious errors or lacking of specificity (higher variance pairs).

### 4.3.2 Variance vs. Quality Tags: Preference Signals Outperform Standalone Judgments

While LLM-based method like InsTag (Lu et al., 2023) filters instructions via LLM-as-a-Quality-Judge, we demonstrate that preference-driven variance filtering better aligns with preference learning.

**Setup.** We compare: (1) **Variance-Based**: Low-variance instructions ($v_i \leq 1.5$) from Section 4.3.1; (2) **InsTag**: Top-quality instructions (score=5) jointly tagged by Llama-3.1-70B-Instruct and Qwen2.5-72B-Instruct (see Appendix A.1.2) while controlling other settings.

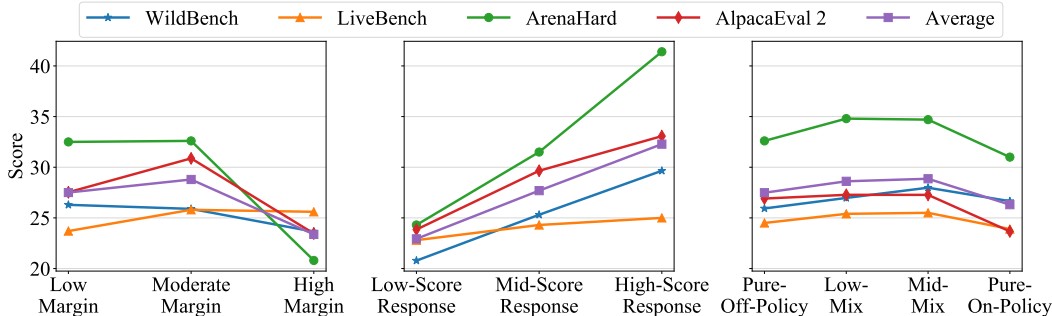

Figure 4: Performance across different benchmarks, comparing different relative score margins (**left**), absolute score thresholds (**middle**), and on/off-policy mixing strategies (**right**). We prioritize pairs with moderate score margin and higher absolute score, while ensuring hybrid mixing of on/off-policy responses.

**Results.** Our variance-based instruction selection method achieves a +2.2 average improvement over instructions tagged as high-quality by InsTag, as shown in Figure 3 (left). This holds even when InsTag uses two annotators for comprehensive evaluation. This indicating that leverage the lateral preference signals outperform standalone judgements.

### 4.3.3 *Multi-Turn vs. Single-Turn Context Analysis*

Having identified low-variance instructions as critical for alignment performance, we extend our analysis to instruction structure—specifically, whether multi-turn conversational contexts improve preference learning. While variance-based filtering prioritizes task specificity, multi-turn instructions may enhance models' ability to handle dialog-specific nuances. To test this, we split single-turn and multi-turn contexts instructions, constructing the preference pairs the same way as before.

**Results.** Multi-turn context instructions yield marginal gains on MT-Bench Turn 2 (+0.7), with no improvement on single-turn benchmarks, as shown in Figure 3 (right). This suggests that multi-turn instructions slightly enhances chat-specific capabilities, while current benchmarks underemphasize multi-turn alignment, implying that while multi-turn instructions show niche utility, their value hinges on future benchmarks prioritizing conversational depth over single-turn tasks.

### 4.4 Response Pair Construction

Effective response pair construction must balance three competing objectives: (1) **Signal Clarity** (contrasts should be unambiguous), (2) **Response Quality** (both responses must be competent to avoid trivial comparisons), and (3) **Policy Alignment** (mix on/off-policy responses to avoid distribution shift). Prior work often optimizes one dimension at the expense of others—we systematically address all three through:

> 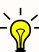 **Takeaway 3**: Optimal pairs combine **moderate margins** ($\Delta = 2$ *or* 3) for clear contrasts, **high absolute scores** ($s(x, y_w) \geq 8$) for quality, and **hybrid mixing** (one on-policy + one off-policy response) for diversity—establishing a balanced approach for effective preference modeling.

### 4.4.1 *Relative Score Margin: Balancing Noise and Signal*

While DPO in Equation (3) treats all preference pairs equally, we find a moderate score margin ($\Delta = 2$ *or* 3) optimizes alignment performance.

**Setup.** Using Llama-3.1-70B-Instruct and Single-Basic annotation, we construct pairs $(y_w, y_l)$ with low ($\Delta = 1$), moderate ($\Delta = 2$ *or* 3), and high ($\Delta \geq 4$) margins, balancing dataset size across groups. In order to keep enough preference pairs under different settings, we choose split thresholds based on score margin distribution in Appendix A.4.

**Results.** Moderate margin achieves higher (+1.29/+5.42) average performance than low/high margins, as shown in Figure 4 (left), excelling in instruction following (AlpacaEval 2: +3.35/+7.42) and reasoning (ArenaHard: +0.2/+11.8) compared to high margin setting.

This is because a moderate margin provides clear preference signals without oversimplifying the learning objective, avoiding noise (low $\Delta$) or overfitting (high $\Delta$).

### 4.4.2 *Absolute Score Threshold: Prioritizing High-Quality Responses*

Given that the moderate relative score margin matters, here we want to investigate the effect of absolute score thresholds. We find that preference pairs with higher absolute scores (e.g., 9 vs. 7) outperform pairs with identical margins but lower scores (e.g., 3 vs. 1), emphasizing the importance of overall response quality.

**Setup.** We construct preference pairs $(y_w, y_l)$ for instruction $x$ based on the absolute score of chosen response $s(x, y_w)$ with controlled relative score margin distribution. We defined three settings: (1) High: $s(x, y_w) \geq 8$; (2) Mid: $s(x, y_w) = 7$; (3) Low: $s(x, y_w) \leq 6$. We choose the threshold based on our observation of the score distribution, detailed in Appendix A.4.

**Results.** High-score pairs achieve the most significant performance across all benchmarks, leading a +9.35 compared to low-score pairs overall, as shown in Figure 4 (middle), with the largest gains in instruction following (AlpacaEval 2: +9.22) and reasoning (ArenaHard: +17.1). This indicates that while DPO optimizes relative preferences, high-score pairs provide clearer learning signals, as both responses are competent but distinguishable. Low-score pairs risk amplifying noise from inherently poor responses.

### 4.4.3 *On/Off-Policy Mixing: Diversity for Effective Contrast*

There is a line of research validating the effectiveness of on-policy samples in preference learning (Dubey et al., 2024; Lambert et al., 2024a). However, how to best mix the on/off policy samples remain uncertain. Here we investigate the effect of different mix strategies.

**Setup.** For each instruction, we annotate 4 on-policy (Llama-3.1-Tulu-3-8B-SFT) and 4 off-policy responses (external models). We compare 4 mixing strategies based on the mix rate of on-policy samples: (1) **Pure-Off**: Pairs from off-policy only (totally $C_4^2 = 6$ pairs); (2) **Low-Mix**: Pairs from 5 responses including 1 on-policy and 4 off-policy (totally $C_{4+1}^2 = 10$ pairs); (3) **Mid-Mix**: 1 on-policy paired with every off-policy (totally 4 pairs); (4) **Pure-On**: Pairs from on-policy only (totally $C_4^2 = 6$ pairs). All strategies sample 4 pairs for each instruction with identical margin distribution.

**Results.** Mid-Mix strategy yields the best result across all benchmarks compared to Pure-Off (+1.38) and Pure-On (+2.56) (Figure 4, right). This indicates that a moderate mix rate of on/off-policy responses ensures every DPO update contrasts the current policy with external responses and avoids overfitting to static datasets while maintaining policy relevance.

## 4.5 Combined Impact

To quantify the synergistic impact of our findings, we incrementally integrate optimal components into a baseline DPO pipeline, measuring performance gains stepwise. Starting from Tulu 3 (SFT), we build progressively while maintaining identical hyperparameters and dataset size (14k pairs):

- **Baseline**: DPO with vanilla preference data using Pair-Basic annotation (wo. margin/quality control).
- **+Annotations**: Change to Single-Basic strategy.
- **+Response Pairs**: Moderate margins and high absolute scores, with Mid-Mix on/off-policy pairs.
- **+Instructions**: Leverage low-variance filtering for instruction selection.

**Results.** Cumulative integration yields +5.3 average performance gain steadily, as shown in Figure 1 (right). The change into Single-Basic strategy and higher absolute response scores showing the significant improvements (+1.78, +1.6); detailed results for each benchmark can refer to Appendix C.2. Since our experiment only leverages 14k pairs, we believe the combined impact will continue to scale. The size of preference datasets in previous state-of-the-art models (Dubey et al., 2024; Lambert et al., 2024a) is far larger than this.

### 4.6 Cross-Verification

To validate the robustness of our findings, we replicate key experiments using the instructions from UltraFeedback dataset and Qwen2.5-72B-Instruct as annotator. Results remain consistent across both configurations (see Appendix C.3), confirming the generalizability of our **AIR** framework's principles under varying instruction sources and annotator capabilities. This consistency underscores that our insights reflect intrinsic dataset design dynamics, not implementation-specific artifacts.

## 5 Conclusion

In this work, we propose **AIR**, a framework to systematically dissect preference datasets into three core components—**A**nnotations, **I**nstructions, and **R**esponse Pairs—and quantify their alignment impact. Through systematic experiments, we establish that streamlined annotation with generative scoring, low-variance instruction selection, and response pairs balancing score margins and absolute quality collectively elevate alignment performance. These principles shift dataset design from ad hoc scaling to component-aware optimization, enabling practitioners to build reproducible, high-quality datasets. Future work includes extending **AIR** to larger-scale datasets and diverse algorithms with dynamic component interactions. By bridging dataset design and alignment outcomes, our framework advances preference learning through interpretable, data-centric innovation.

## Acknowledgments

This work was supported by Beiing Advanced Innovation Center for Future Blockchain and Privacy Computing, Alibaba Group through Alibaba Innovative Research Program and Shanghai Artificial Intelligence Laboratory.

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

## A Dataset Details

### A.1 Instructions

#### A.1.1 Instruction Datasets

We utilized two datasets: GPT-4-generated ShareGPT[1] and UltraFeedback[2]. Here, we provide a detailed overview of each dataset.

**ShareGPT.** It comprises a multilingual corpus of multi-turn conversational exchanges curated from high-quality GPT-4 interactions. It contains 103,415 entries across three subsets: 6,206 refined GPT-4 dialogues focusing on knowledge-intensive tasks (e.g., reasoning, programming, and multilingual translation), 58,674 entries reformatted from ShareGPT-V3 with informal multilingual dialogues, and 38,535 Chinese-specific exchanges addressing technical queries and creative tasks. The dataset features conversations in Chinese (both simplified and traditional), English, Japanese, and Korean, with each entry structured as sequential human-AI interaction pairs. In this paper, we use its subset with 58,674 entries reformatted from ShareGPT-V3.

**UltraFeedback.** The UltraFeedback dataset represents a large-scale preference collection framework designed for training reward models and critical evaluators. It aggregates 63,967 prompts from six heterogeneous sources including UltraChat, ShareGPT, and domain-specific QA datasets, generating 256k responses through diverse LLMs ranging from commercial systems (GPT-4, GPT-3.5 Turbo) to open-source models (LLaMA variants, Falcon, MPT). Each prompt elicits four responses annotated by GPT-4 across four dimensions: instruction adherence, truthfulness, honesty, and helpfulness. The dataset's design emphasizes diversity through stratified sampling of instructions and heterogeneous model selection, coupled with fine-grained numerical and textual feedback annotations. In this paper, we use all its instructions.

ShareGPT4 conversations were filtered by a 4096-token maximum length constraint and concatenated across dialogue turns, yielding 33,717 refined instructions. UltraFeedback underwent a basic 4096-token length filtration to ensure response consistency.

---

[1] https://huggingface.co/datasets/shibing624/sharegpt_gpt4
[2] https://huggingface.co/datasets/openbmb/UltraFeedback

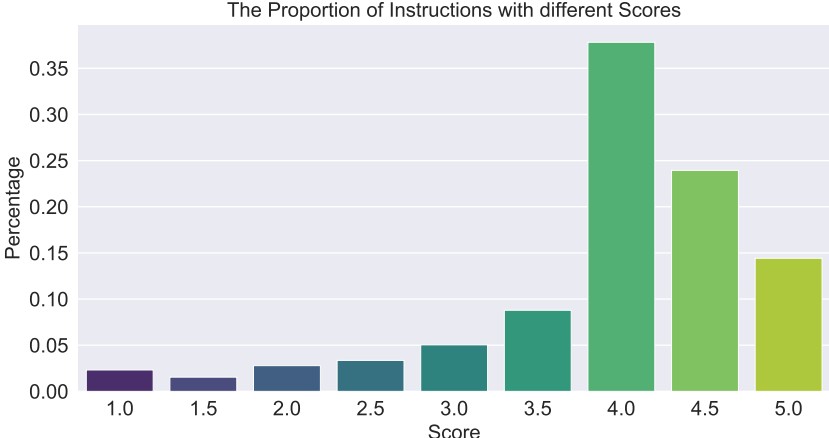

Figure 5: Score distribution of instructions evaluated by Llama-3.1-70B-Instruct and Qwen2.5-72B-Instruct (averaged).

### A.1.2  InsTag Details

As mentioned in Section 4.3.2, inspired by Instag(Lu et al., 2023), the quality of instructions used for preference learning can be evaluated by LLM-as-a-Quality-Judge. Specifically, we utilize two high-performing open-source LLMs - Llama-3.1-70B-Instruct and Qwen2.5-72B-Instruct - to score the instructions on a 1-5 scale, then calculate the average of the two scores as the final evaluation. The distribution of average scores is shown in Figure 5.

The prompt for instruction quality evaluation is as follows:

---

**Prompt for Evaluating Instruction Quality**

You are an expert evaluator tasked with rating the quality of the single or multi-turn dialogue based on its clarity, specificity, and coherence.

## The rating scale is as follows:
— Very Poor: The dialogue is unclear, vague, or incoherent. It lacks essential information and context.
— Poor: The dialogue is somewhat unclear or lacks important details. It requires significant clarification.
— Average: The dialogue is moderately clear and specific. It may require some additional information for a complete understanding.
— Good: The dialogue is clear, specific, and mostly well-formed. It provides sufficient context for understanding the user's intent.
— Excellent: The dialogue is very clear, specific, and well-articulated. It contains all the necessary information and context for providing a comprehensive response.

## Conversation History
{conversation_history}

Please directly evaluate the overall quality of the response by 5 levels in the following format:
Dialog quality: [Very Poor/Poor/Average/Good/Excellent]

---

### A.2  Responses

Our experiments leverage 17 open-source LLMs (Table 2) spanning diverse architectural families:

| Model Family | Model Name | Avg. Token Length | Avg. Score |
|---|---|---|---|
| Llama | Meta-Llama-3.1-8B-Instruct | 440.67 | 6.95 |
| | Meta-Llama-3.1-70B-Instruct | 431.78 | 7.13 |
| | Llama-2-13b-chat-hf | 405.27 | 6.59 |
| | Llama-3.1-Tulu-3-8B-SFT | 317.62 | 6.69 |
| Gemma | gemma-2-2b-it | 478.53 | 6.90 |
| | gemma-2-9b-it | 375.03 | 7.11 |
| | gemma-2-27b-it | 1002.39 | 6.14 |
| Phi-3 | Phi-3-mini-128k-instruct | 588.54 | 3.59 |
| | Phi-3-small-128k-instruct | 366.03 | 6.80 |
| | Phi-3-medium-128k-instruct | 492.73 | 6.11 |
| Qwen | Qwen2-1.5B-Instruct | 283.18 | 5.45 |
| | Qwen2-7B-Instruct | 375.74 | 6.85 |
| | Qwen2-72B-Instruct | 350.10 | 7.05 |
| Mistral | Mixtral-8x7B-Instruct-v0.1 | 327.91 | 6.84 |
| | Mistral-7B-Instruct-v0.3 | 343.15 | 6.84 |
| Others | DeepSeek-V2-Lite-Chat | 319.52 | 6.54 |
| | MiniCPM3-4B | 260.63 | 6.37 |

Table 2: Overview of LLMs in the model pool: model families, average token lengths of responses, and average annotation scores.

- **Llama Family (Meta & AI2)**: Includes parameter-scaled variants (8B, 70B) with enhanced tokenization and grouped query attention (GQA): Meta-Llama-3.1-8B-Instruct, Meta-Llama-3.1-70B-Instruct, Llama-2-13b-chat-hf, Llama-3.1-Tulu-3-8B-SFT.
- **Gemma Family (Google)**: Lightweight models (2B–27B) optimized for dialogue and instruction following: gemma-2-2b-it, gemma-2-9b-it, gemma-2-27b-it.
- **Phi-3 Family (Microsoft)**: Compact yet powerful models with 128k context windows, employing block expansion and quantization-aware training: Phi-3-mini-128k-instruct, Phi-3-small-128k-instruct, Phi-3-medium-128k-instruct.
- **Qwen2 Family (Alibaba)**: Scalable models (1.5B–72B) with sliding window attention for efficient long-context processing: Qwen2-1.5B-Instruct, Qwen2-7B-Instruct, Qwen2-72B-Instruct.
- **Mistral Family (Mistral AI)**: High-performance architectures, including the mixture-of-experts (MoE) Mixtral-8x7B-Instruct-v0.1 and its dense counterpart Mistral-7B-Instruct-v0.3.
- **Specialized Models**: DeepSeek-V2-Lite-Chat: Hybrid attention mechanisms for efficiency; MiniCPM3-4B: Compact 4B model with multilingual support.

Here is a generation prompt example:

---

**Generation Prompt Example**

`[INST]` Summarize the main ideas of Jeff Walker's Product Launch Formula into bullet points as it pertains to a growth marketing agency implementing these strategies and tactics for their clients... `[/INST]`
Well, I'm quite partial to a good squeeze of fresh lemon juice. It adds just the right amount of zesty flavour to whatever I'm cooking up in the kitchen!``
`[INST]` Summarize the main ideas of Brendon Burchard's Experts Academy into bullet points as it pertains to a growth marketing agency implementing these strategies and tactics for their clients... `[/INST]`
Well, I'm quite partial to a good squeeze of fresh lemon juice. It adds just the right amount of zesty flavour to whatever I'm cooking up in the kitchen!``

---

> [INST] What are the mental triggers in Jeff Walker's Product Launch Formula and "Launch" book? [/INST]

## A.3 Annotations

### A.3.1 Annotator Models

In this study, we utilize three annotator models to assess and refine the quality of generated responses. These include a classifier-based model, Skywork-Reward-Gemma-2-27B-v0.2[1], and two generative models, Llama-3.1-70B-Instruct[2] and Qwen-2.5-72B-Instruct[3]. Here we report their score on RewardBench[4] in Appendix A.3.1, illustrating their alignment capabilities and reliability in assessing response quality.

| Model | Score | Rank |
|---|---|---|
| Skywork-Reward-Gemma-2-27B-v0.2 | 94.3 | 1 |
| Llama-3.1-70B-Instruct | 84.0 | 40 |
| Qwen-2.5-72B-Instruct | 82.3[*] | 46[*] |

Table 3: RewardBench (Lambert et al., 2024b) scores and leaderboard rankings. [*]Self-reported score (not officially listed on leaderboard).

Scores reflect model performance at the time when conducting the experiments, with Qwen-2.5-72B-Instruct's score independently verified through our evaluation.

### A.3.2 Annotation Strategies: Prompt Design for Response Scoring

We evaluate six annotation strategies for prompt design using generative models, considering three key dimensions: **response count**, **guideline complexity**, and **explanation requirements**. By combining different levels within each dimension, we design a total of six distinct prompt strategies, ensuring a comprehensive assessment of annotation methods.

**Response Count.**   This dimension determines whether responses are evaluated individually or in a comparative manner.

- **Single:** The model assigns a score to one response at a time, without explicit comparison to alternatives.
- **Pairwise Comparison:** The model evaluates two responses or more simultaneously, requiring a relative assessment of their quality.

**Guideline Complexity.**   Guidelines provide evaluators with explicit criteria to structure their assessments.

- **w/o. Guidelines:** The model assigns scores purely based on its internal judgment without external evaluation criteria.
- **w. Coarse Guidelines:** Basic scoring instructions are provided, ensuring a minimal level of consistency in evaluation.
- **w. Fine-Grained Guidelines:** Fine-grained evaluation incorporates task-specific preference questions, generated based on the user query, to assess response quality in a structured manner.

---

[1] https://huggingface.co/Skywork/Skywork-Reward-Gemma-2-27B-v0.2
[2] https://huggingface.co/meta-llama/Llama-3.1-70B-Instruct
[3] https://huggingface.co/Qwen/Qwen2.5-7B-Instruct
[4] https://huggingface.co/spaces/allenai/reward-bench

**Explanation Requirements.** This dimension controls whether the model justifies its scoring decisions.

- **w/o. Explanation:** The model outputs only a numerical score without explaining its reasoning.
- **w. Explanation:** The model first provides a reasoning process before assigning a final score.

By systematically combining these dimensions, we construct six distinct annotation prompts, ranging from minimal (Single-Basic: scoring one response without guidelines or explanation) to highly structured (Pair-Guided-Explained-Fine-Grained: comparing two responses using fine-grained guidelines and detailed explanations). Below, we describe each strategy in detail.

**Single-Basic.** This is the simplest annotation strategy, where the model evaluates each single response without any additional guidelines or explanations.

---

**Prompt for Single-Basic Annotation Strategy**

You are an expert evaluator tasked with providing a simple numerical score for the response given the conversation history and user query.

## Conversation History
<|begin_history|>
{dialogue_history}
<|end_history|>

## Current User Query
<|begin_query|>
{instruction}
<|end_query|>

## Response to Evaluate
<|begin_response|>
{response}
<|end_response|>

Please provide your evaluation by directly scoring the overall quality of the response from 0 to 9 in the following format:

SCORE: [0-9]

---

**Pair-Basic.** For pairwise comparisons, this strategy asks the model to assign independent scores to two responses without any additional explanations.

---

**Prompt for Pair-Basic Annotation Strategy**

You are an expert evaluator tasked with comparing two responses to the same query.

## Conversation History
<|begin_history|>
{dialogue_history}
<|end_history|>

## Current User Query
<|begin_query|>
{instruction}
<|end_query|>

---

## Response A
<|begin_response_a|>
{response_a}
<|end_response_a|>

## Response B
<|begin_response_b|>
{response_b}
<|end_response_b|>

Please provide your evaluation by directly scoring the overall quality of the responses from 0 to 9 in the following format exactly, where 0 is the worst and 9 is the best.

Your response MUST follow this exact format:
SCORE_A: [0–9]
SCORE_B: [0–9]

**Pair-Explained.**  A pairwise evaluation strategy where the model first highlights the differences between two responses before assigning scores, improving transparency in preference selection.

**Prompt for Pair-Explained Annotation Strategy**

You are an expert evaluator tasked with providing a simple numerical score for the response given the conversation history and user query.

## Conversation History
<|begin_history|>
{dialogue_history}
<|end_history|>

## Current User Query
<|begin_query|>
{instruction}
<|end_query|>

## Response A
<|begin_response_a|>
{response_a}
<|end_response_a|>

## Response B
<|begin_response_b|>
{response_b}
<|end_response_b|>

Please provide your evaluation by first pointing out the pros and cons of both responses without polite phrases as short as you can, ensuring that the order of the responses does not affect your judgment. Then rate the overall quality of the responses from 0 to 9, with 0 being worst and 9 being best.
Your response MUST follow this exact format:

EXPLANATION:
[Your detailed comparison of both responses]

SCORE_A: [0–9]
SCORE_B: [0–9]

**Pair-Guided.** This is a comparison strategy where predefined scoring guidelines are provided to help the model evaluate and score two responses independently without explanation. The Rough scoring guidelines are as follows:

- **8–9**: Exceptional response that excels in all aspects.
- **6–7**: Strong response with minor room for improvement.
- **4–5**: Adequate response with some notable gaps.
- **2–3**: Poor response with significant issues.
- **0–1**: Severely inadequate or irrelevant response.

---

**Prompt for Pair-Guided Annotation Strategy**

You are an expert evaluator tasked with providing a simple numerical score for the response given the conversation history and user query.

## Conversation History
<|begin_history|>
{dialogue_history}
<|end_history|>

## Current User Query
<|begin_query|>
{instruction}
<|end_query|>

## Response A
<|begin_response_a|>
{response_a}
<|end_response_a|>

## Response B
<|begin_response_b|>
{response_b}
<|end_response_b|>

Scoring Guidelines:
– 8–9: Exceptional response that excels in all aspects
– 6–7: Strong response with minor room for improvement
– 4–5: Adequate response with some notable gaps
– 2–3: Poor response with significant issues
– 0–1: Severely inadequate or irrelevant response

Please provide your evaluation by directly scoring the overall quality of the responses from 0 to 9 in the following format exactly, where 0 is the worst and 9 is the best.

Your response MUST follow this exact format:
SCORE_A: [0–9]
SCORE_B: [0–9]

---

**Pair-Guided-Explained.** This is a combination of Pair-Guided and Pair-Explained, requiring the model to follow scoring guidelines and explicitly articulate the reasoning behind its comparison before assigning scores.

---

**Prompt for Pair-Guided-Explained Annotation Strategy**

You are an expert evaluator tasked with providing a simple numerical score for the response given the conversation history and user query.

---

## Conversation History
<|begin_history|>
{dialogue_history}
<|end_history|>

## Current User Query
<|begin_query|>
{instruction}
<|end_query|>

## Response A
<|begin_response_a|>
{response_a}
<|end_response_a|>

## Response B
<|begin_response_b|>
{response_b}
<|end_response_b|>

Scoring Guidelines:
– 8–9: Exceptional response that excels in all aspects
– 6–7: Strong response with minor room for improvement
– 4–5: Adequate response with some notable gaps
– 2–3: Poor response with significant issues
– 0–1: Severely inadequate or irrelevant response

Please provide your evaluation by first point out the pros and cons of both responses without polite phrases as short as you can, ensuring that the order of the responses does not affect your judgment. Then rate the overall quality of the responses from 0 to 9, with 0 being worst and 9 being best.
Your response MUST follow this exact format:

EXPLANATION:
[Your detailed evaluation based on the scoring guidelines]

SCORE_A: [0–9]
SCORE_B: [0–9]

**Pair-Guided-Explained-Fine-Grained.** This is the most detailed annotation strategy, incorporating task-specific preference questions in the pairwise evaluation. The model must address each question for both responses before scoring. The prompt includes:

**Prompt for Pair-Guided-Explained-Fine-Grained Annotation Strategy**

You are an expert evaluator tasked with providing a simple numerical score for the response given the conversation history and user query.

## Conversation History
<|begin_history|>
{dialogue_history}
<|end_history|>

## Current User Query
<|begin_query|>
{instruction}
<|end_query|>

## Response A
<|begin_response_a|>
{response_a}
<|end_response_a|>

## Response B
<|begin_response_b|>
{response_b}
<|end_response_b|>

Please evaluate both responses by first answering the following task-specific preference questions with Yes/No followed by a brief explanation. Then rate the overall quality of the responses from 0 to 9, with 0 being worst and 9 being best. Avoid polite phrases and be as concise as possible. The order of responses should not affect your judgment.

## Task–specific Preference Questions
{questions_prompt}

Your response MUST follow this exact format:

EXPLANATION:
Response A:
{explanation_prompt}

Response B:
{explanation_prompt}

[Overall comparison explanation considering the fine–grained evaluation results]

SCORE_A: [0–9]
SCORE_B: [0–9]

Your scoring should be based on how many task-specific questions were answered with "Yes" and the quality of fulfillment for each criterion.

For fine-grained annotation, we first use Llama-3.1-8B-Instruct to generate task-specific preference questions based on the user query. These questions are then integrated into the fine-grained annotation prompt to guide the evaluation process. The annotation model responds to each preference question with a "Yes" or "No", accompanied by a brief explanation, before assigning a final quality score. The explanation format follows "[Yes/No] - [Brief explanation]" and is repeated for each preference question. The prompt template to generate preference questions is as follows:

**Prompt for Generate Fine-Grained Preference Questions**

You are a task analyzer that examines conversations and queries to identify task categories and generate relevant preference questions. Your analysis helps evaluate if responses meet user requirements.

RULES:
1. Analyze both the conversation context (if any) and final query to understand the complete task
2. Category should be brief (2-4 words) but precisely describe the task type
3. Generate 4-7 preference questions that:
- Focus on whether the response directly addresses user's specific request
- Check if all parts of the user's query are fulfilled
- Verify if the response provides exactly what was asked for

- Consider any context from previous conversation turns
4. Questions should help verify if a response completely satisfies what the user asked for

IMPORTANT: You MUST respond ONLY with a valid JSON object in exactly this format:

```
{{
    "category": "brief task description",
    "preference_questions": [
        "specific question 1?",
        "specific question 2?",
        "specific question 3?",
        "..."
    ]
}}
```

Do not include any other text or explanation.

Below is an example for generated preference questions:

### Example for Generated Preference Questions

**User Query:** Summarize the main ideas of Jeff Walker's Product Launch Formula into bullet points as it pertains to a growth marketing agency implementing these strategies and tactics for their clients...

**category:** product summary
**preference questions:**
- Does the response provide a clear summary of the main ideas of Jeff Walker's Product Launch Formula?
- Are the bullet points specifically tailored for a growth marketing agency implementing these strategies for clients?
- Does the response accurately convey the key takeaways from the Product Launch Formula for this context?
- Are the bullet points concise and easy to understand?
- Does the response cover all the essential aspects of the Product Launch Formula relevant to growth marketing agencies?

These fine-grained preference questions serve as approximately objective criteria for evaluating response quality, ensuring that annotations capture meaningful distinctions in model outputs.

We also analyze the correlation between the proportion of "Yes" answers to preference questions and the overall response score. We conduct experiments on two datasets, *ShareGPT-v3* and *UltraFeedback*, using scatter plot distributions to analyze the relationship between preference question agreement and final scores. As shown in Figure 6. there is a clear positive correlation between the proportion of "Yes" answers and the assigned score—responses with higher agreement on preference questions consistently receive higher scores. Conversely, we observe a negative correlation with the proportion of "No" answers, indicating that responses failing to meet preference criteria tend to receive lower scores.

### A.3.3 Scoring Methods

Here we introduce the detailed calculation of three scoring methods:

**Single-Basic.** In this method, a single inference run is performed using greedy decoding. The score is assigned based solely on the model's deterministic response. Since no randomness is introduced in this approach, it ensures high consistency across repeated evaluations.

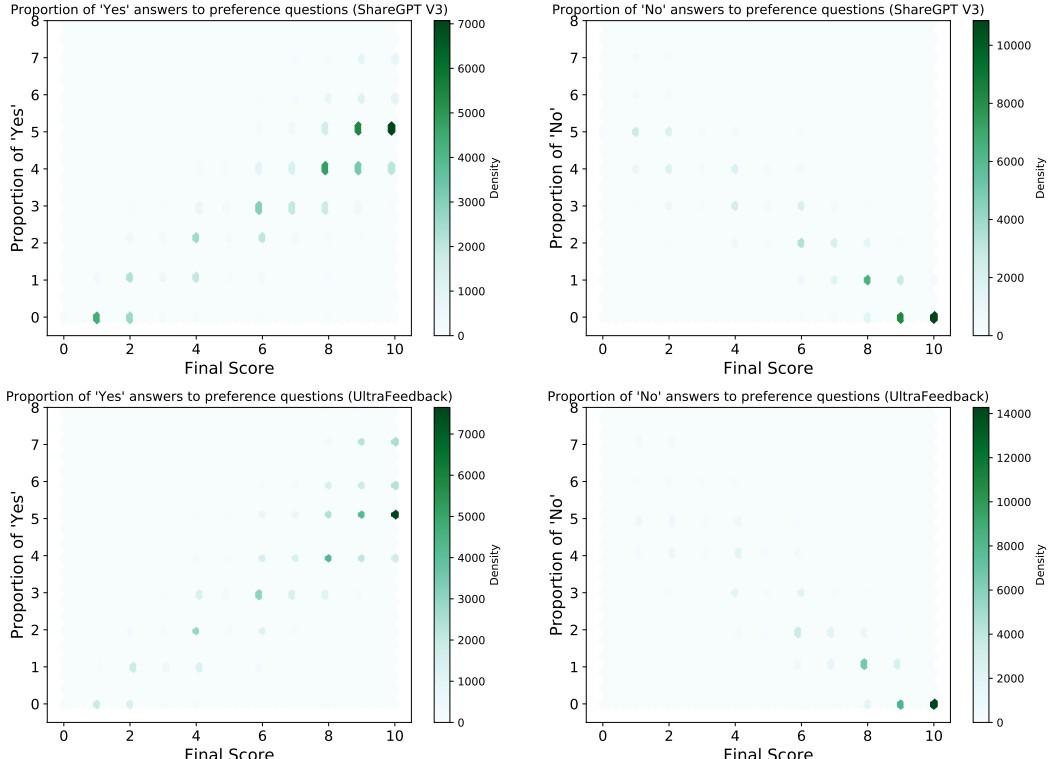

Figure 6: Correlation between preference question agreement and final score. A clear positive correlation between the proportion of "Yes" answers and a negative correlation with the proportion of "No" answers can be observed from both ShareGPT and UltraFeedback datasets.

**Single-Avg.** The final score in this method is computed by averaging the scores from five independent inference runs, each performed with a temperature setting of 1.0. The final score is calculated as the average of the individual scores:

$$S_{\text{avg}} = \frac{1}{N} \sum_{i=1}^{N} S_i, \quad N = 5 \tag{4}$$

where $S_i$ represents the score from the $i$-th run.

**Single-Prob.** This method calculates the final score by considering the probability distribution over possible score tokens (from 0 to 9) predicted by the model. During inference, the model outputs raw scores (logits) for each possible score. These logits are then converted into probabilities using the softmax function. The final score is obtained by weighting each score by its corresponding probability.

We let $\text{logit}_k$ represent the raw output score for score $k$. The corresponding probability $P_k$ is computed using the softmax function:

$$P_k = \frac{e^{\text{logit}_k}}{\sum_{i=0}^{9} e^{\text{logit}_i}}, \quad k = 0, 1, 2, \dots, 9 \tag{5}$$

Then, the final score $S_{\text{prob}}$ is computed by summing the weighted probabilities of each score:

$$S_{\text{prob}} = \sum_{k=0}^{9} k P_k \tag{6}$$

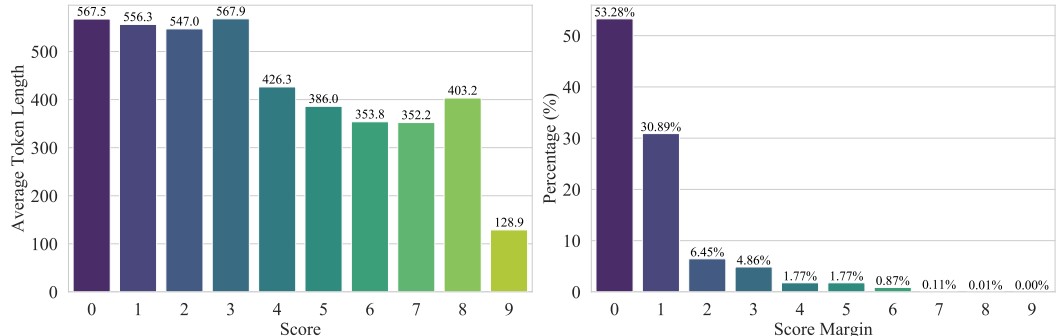

Figure 7: **Left:** Average Token Length by Score. **Right:** Score Margin Distribution.

| Model | Batch Size | Max Length | Learning Rate | Beta | Epoch | LR Scheduler |
|---|---|---|---|---|---|---|
| Llama-3.1-Tulu-3-8B-SFT | 16 | 4096 | 5e-7 | 0.1 | 1 | Cosine |

Table 4: Hyperparameters for DPO training.

where $P_k$ is the predicted probability for score $k$, and $k$ is the score value (ranging from 0 to 9). This method captures the model's confidence in each possible score and is particularly useful in cases where the model's preferences are uncertain or borderline.

## A.4 Final Dataset Statistics

**Token Length and Average Scores.** Table 2 presents the token length distributions and average annotation scores using Single-Basic strategy for each model in our dataset, calculated using the Llama-3 tokenizer to ensure consistency.

**Length Distribution Across Scores.** We analyze the correlation between response token length and assigned scores (0-9). Figure 7 (Left) shows the average token length for each score, indicating no significant length bias in annotation.

**Score Margin Distribution.** The distribution of score margins is presented in Figure 7 (Right), where margins of 0 and 1 are the most common, followed by larger margin frequencies tapering off.

**Annotation Consistency Across Annotation Strategies.** The consistency of preference annotations across different strategies is visualized in Figure 8. The results demonstrate high consistency between those annotation methods.

**Variance distribution for multiple responses per instruction.** In Section 4.3.1, we calculate variances of scores across different responses to the same instruction. The distribution of the variances is shown in Figure 9, where we observe a long-tailed distribution of instruction variances. Our experimental results demonstrate that using instructions with lower variance yields better DPO training outcomes.

# B Experiments Setup

## B.1 Training Details

For preference learning, we present some key hyperparameters related to DPO in Table 4. Additionally, we utilize the OpenRLHF framework (Hu et al., 2024) to conduct full-parameter DPO of Llama-3.1-Tulu-3-8B-SFT on 8 80GB A800s. All of our DPO experiments utilize these hyper-parameters consistently.

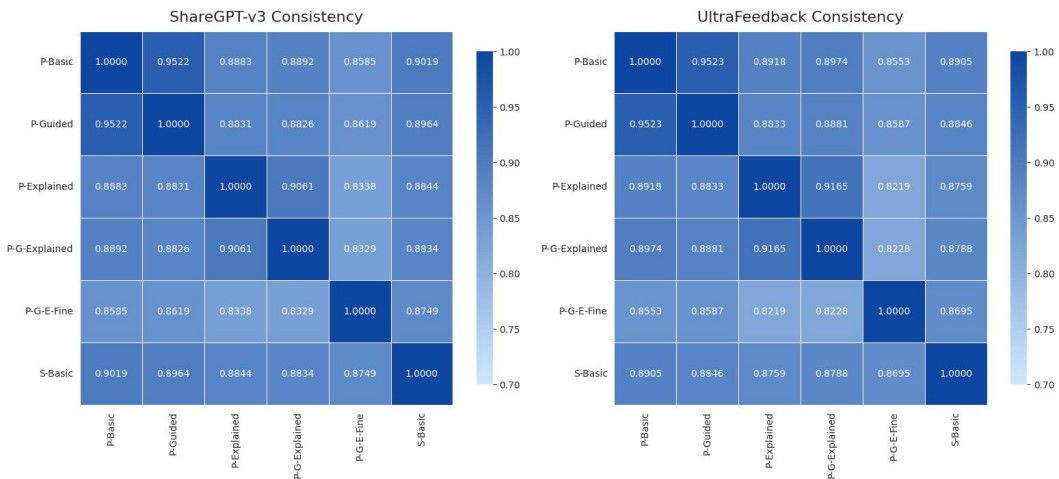

Figure 8: Annotation consistency across annotation strategies. Left: ShareGPT; Right: UltraFeedback.

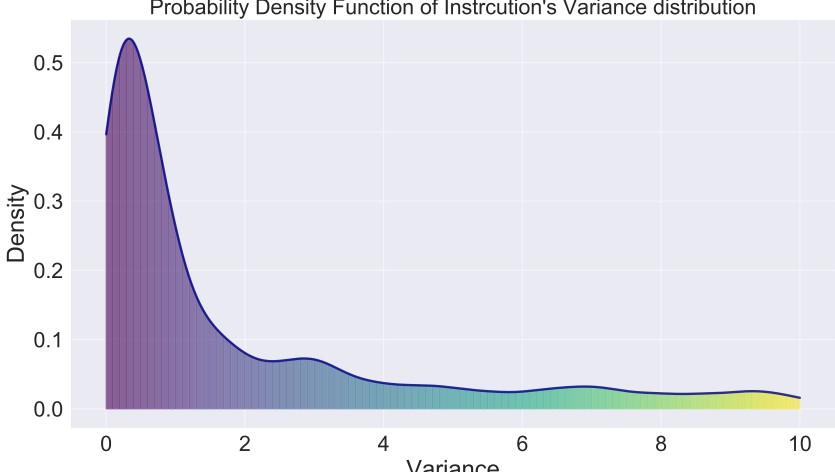

Figure 9: Score variance distribution for variance-based method.

## B.2 Evaluation Details

This section provides detailed descriptions of the evaluation benchmarks used in our experiments, including the capabilities assessed, evaluation methodologies, and primary metrics.

**MTBench (Zheng et al., 2023b).** MTBench evaluates instruction-following and conversational capabilities through diverse, multi-turn dialogue prompts covering tasks such as math, reasoning, writing, and coding. The benchmark consists of 80 test examples. Evaluations are subjective, using GPT-4o as the judge model to score responses from 1 to 10 based on helpfulness, correctness, coherence, and related aspects.

**ArenaHard (Li et al., 2024b).** ArenaHard consists of 500 challenging crowd-sourced conversational prompts specifically designed to stress test advanced reasoning, complex instruction follow-up, and nuanced knowledge retrieval capabilities. Evaluations are subjective and performed using the LLM-as-a-Judge method, with GPT-4-0314 serving as both the judge and the baseline model. The primary metric is the Elo rating derived from pairwise comparisons.

**AlpacaEval 2.0 (Dubois et al., 2024).** AlpacaEval 2.0 assesses instruction-following accuracy and general-purpose alignment using 805 user-generated instructions spanning general

knowledge, reasoning, writing, and coding tasks. Evaluations are subjective, using GPT-4 Preview (11/06) as both the baseline and the auto-annotator. The main metric is length-controlled (LC) win rate, which alleviates length biases inherent in GPT-4 evaluations.

**WildBench (V2) (Lin et al., 2024).** WildBench (V2) benchmarks model performance using 1,024 challenging real-world user queries selected from extensive human-chatbot interactions. Evaluations systematically use task-specific checklists and provide structured explanations to justify scores and comparisons. Judges can vary; we specifically used GPT-4o and primarily employed WB-Score, which individually assesses response quality rapidly and cost-effectively.

**Eurus Evaluation Suite (Yuan et al., 2024).** The Eurus evaluation suite rigorously tests model capabilities across coding, mathematics, logical reasoning, and instruction-following tasks using carefully curated, expert-generated prompts. It includes a diverse set of benchmarks: HumanEval, LeetCode, and MBPP for coding; GSM-Plus, SVAMP, ASDiv, TheoremQA, and general math evaluations for mathematics; Big Bench Hard (BBH) for reasoning; and IFEval for instruction-following. Evaluations primarily utilize objective methods based on exact match accuracy and computational correctness. Detailed descriptions are available at https://github.com/OpenBMB/Eurus/tree/main/eval.

**LiveBench (White et al., 2024).** LiveBench is a dynamic benchmark designed to mitigate test set contamination by regularly introducing new tasks derived from recently published datasets, arXiv papers, news articles, and IMDb movie synopses. In our experiments, we used the 2024-11-25 version containing 1,136 questions. Evaluations are objective, based on verifiable ground-truth answers.

## C  Addition Experimental Results

### C.1  Further Experiments on On/Off-Policy Mixing

To further investigate on/off-policy responses' effect in preference dataset, based on the **Mid-Mix** configuration in Section 4.4.3 where preference dataset composes of 50% on-policy and 50% off-policy responses, we designed two experimental groups:

**OnChosen-OffReject** - where all chosen responses were high-scoring on-policy responses and all rejected responses were low-scoring off-policy responses;

**OffChosen-OnReject** - the inverse configuration using high-scoring off-policy responses as chosen and low-scoring on-policy responses as rejected.

| Setup | ArenaHard | WildBench | LiveBench |
|-------|-----------|-----------|-----------|
| Mid-Mix | **34.7** | **27.98** | **25.5** |
| OnChosen-OffReject | 27.6 | 22.01 | 20.5 |
| OffChosen-OnReject | 15.9 | 12.43 | 19.3 |

Table 5: Results for **OnChosen-OffReject**, **OffChosen-OnReject** and **Mid-Mix** setups.

The results for these two setups, along with **Mid-Mix**, are presented in Table 5. Notably, while maintaining identical on-policy/off-policy response ratios to the **Mid-Mix** setup, both setups exhibit substantial performance degradation across benchmarks compared to **Mid-Mix**. We hypothesize this stems from the homogeneity of on-policy data originating from a single SFT model. When exclusively employing on-policy data in either chosen or rejected response, LLMs may exploit superficial patterns, thereby learning shortcut heuristics that undermine preference learning efficacy.

Crucially, the **OffChosen-OnReject** setup demonstrates catastrophic alignment collapse. This phenomenon arises from the DPO loss formulation as shown in Equation (3), the alignment process systematically suppresses the likelihood of LLM's own outputs when off-policy responses are designated as chosen while on-policy responses serve as rejected targets.

Such dynamics violate the fundamental assumption of preference alignment, ultimately destabilizing the learning objective.

## C.2 Detailed Results for Combined Experiments

In Figure 1 (right), we demonstrate the performance gains achieved through additive integration of components within our AIR framework. In this section, we quantitatively illustrates the detailed improvements contributed by each integrated element across all benchmarks.

| Setup | MT-Bench | WildBench | Eurus | LiveBench | ArenaHard | Average |
|---|---|---|---|---|---|---|
| *Tulu-3-8B-SFT (Baseline)* | 6.38 | 12.50 | 55.77 | 24.3 | 13.8 | 34.03 |
| w. Vanilla Pref. Data | 6.51 | 25.65 | 56.81 | 24.1 | 30.2 | 40.37 |
| w. Point-Wise Scoring Pref. Data | 6.83 | 26.47 | 57.79 | 24.4 | 33.8 | 42.15 |
| + Moderate score Margin among responses | 7.00 | 25.80 | 57.48 | 25.9 | 33.8 | 42.60 |
| + Higher Absolute Response Score | 7.13 | 29.25 | 57.96 | 25.9 | 35.4 | 43.96 |
| + On/Off-Policy Responses Hybrid Mixed | 7.04 | 31.22 | 58.11 | **26.5** | 36.8 | 44.61 |
| + Inst. with Lower Resp. Score Variance | **7.23** | **32.43** | **58.38** | 26.4 | **38.9** | **45.68** |

Table 6: Details for AIR Framework's Additive improvement across all benchmarks, resulted by stepwise integrating different principles. Scores are normalized to 100-point scale for averaging.

## C.3 Cross-verification

In Section 4, we used Llama-3.1-70B-instruct for the annotation of preferences and used instructions from ShareGPT to generate responses. Through comprehensive experiments, we determined the optimal design for the AIR framework. In this section, we present an analysis of the robustness of the AIR framework from two critical perspectives: the preference annotation model and instruction sources in preference data.

### C.3.1 Robustness Analysis of Annotation Model

To analyze the robustness of annotation model, we use Qwen-2.5-72B-instruct (Yang et al., 2024) for preference annotation while maintaining identical experimental settings to those in Section 4.

Results for annotation strategy are shown in Table 7, which demonstrates the **Single-Basic** configuration consistently outperformed pairwise scoring approaches across all five benchmarks.

We present our experimental results for response pairs in Figure 10, it clearly shows that pairs with moderate score margin, higher absolute score, and mixing on/off-policy responses achieves better results, in align with Llama-based annotation experiments.

The evaluation results for the effect of variances across different response score from the same instruction is shown in Figure 11, we can find that except for Eurus, the low-variance instruction achieves the best performance on all other benchmarks, which is consistent with the conclusions of the main experiments.

### C.3.2 Robustness Analysis of Instruction Sources

To analysis the robustness of instruction sources, we employ instructions in UltraFeedback (Cui et al., 2023) for response generation while maintaining identical experimental settings to those in Section 4.

We present the results for annotation strategy in Table 8, which demonstrates even with the modification in instruction source, the **Single-Basic** setup has the best overall performance across all benchmarks.

| Annotation Strategy | MT-Bench | WildBench | Eurus | LiveBench | ArenaHard | Avg. |
|---|---|---|---|---|---|---|
| *Tulu-3-8B-SFT (Baseline)* | 6.38 | 12.50 | 55.77 | 24.3 | 13.8 | 34.03 |
| Single-Basic | **6.81** | **27.47** | **57.85** | **26.5** | **37.2** | **43.42** |
| Pair-Basic | 6.73 | 26.86 | 57.30 | 21.8 | 33.1 | 41.27 |
| Pair-Guided | 6.67 | 26.92 | 57.21 | 21.5 | 36.6 | 41.79 |
| Pair-Explained | **6.81** | 26.29 | 57.40 | 22.3 | 35.4 | 41.90 |
| Pair-Guided-Explained | 6.62 | 26.21 | 57.40 | 20.7 | 34.7 | 40.04 |
| Pair-Guided-Explained-FG | 6.73 | 24.10 | 56.55 | 23.3 | 32.8 | 40.81 |

Table 7: Impact of 6 annotation strategies on DPO performance. Qwen-2.5-72B-Instruct is employed for preference annotation while maintaining identical experimental setups to those specified in section 4.2.2. FG denotes Fine-Grained. **Single-Basic** shows superior performance, which aligns with the results obtained from Llama-based annotation.

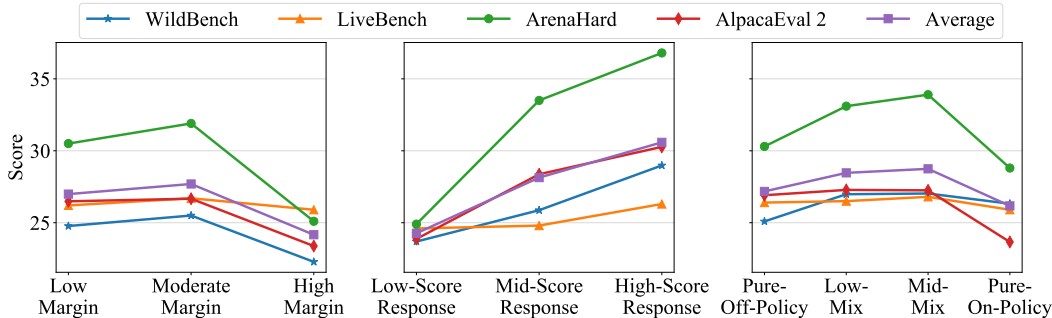

Figure 10: Performance across different benchmarks, comparing different relative score margins (**left**), absolute score thresholds (**middle**), and on/off-policy mixing strategies (**right**). Qwen-2.5-72B-Instruct is employed for preference annotation while maintaining identical experimental setups to those specified in section 4.4. We prioritize pairs with moderate score margin, higher absolute score, and hybrid mixing of on/off-policy responses, which is in align with Llama-based annotation experiments.

| Annotation Strategy | MT-Bench | WildBench | Eurus | LiveBench | ArenaHard | Avg. |
|---|---|---|---|---|---|---|
| *Tulu-3-8B-SFT (Baseline)* | 6.38 | 12.50 | 55.77 | 24.3 | 13.8 | 34.03 |
| Single-Basic | **6.58** | 27.17 | **58.95** | **25.4** | **36.0** | **42.66** |
| Pair-Basic | 6.55 | 24.59 | 58.90 | 23.5 | 34.2 | 41.70 |
| Pair-Guided | 6.44 | 26.56 | 58.38 | 22.4 | 35.0 | 41.19 |
| Pair-Explained | 6.38 | 26.09 | 58.70 | 22.6 | 32.5 | 40.74 |
| Pair-Guided-Explained | 6.56 | **27.43** | 58.30 | 23.4 | 34.9 | 41.93 |
| Pair-Guided-Explained-FG | 6.46 | 23.61 | 58.46 | 24.5 | 31.4 | 40.51 |

Table 8: Impact of 6 annotation strategies on DPO performance. Instructions from Ultrafeedback is employed for response generation while maintaining identical experimental setups to those specified in section 4.2.2. All strategies ensure identical instruction sets and pair counts used for DPO training. FG denotes Fine-Grained.

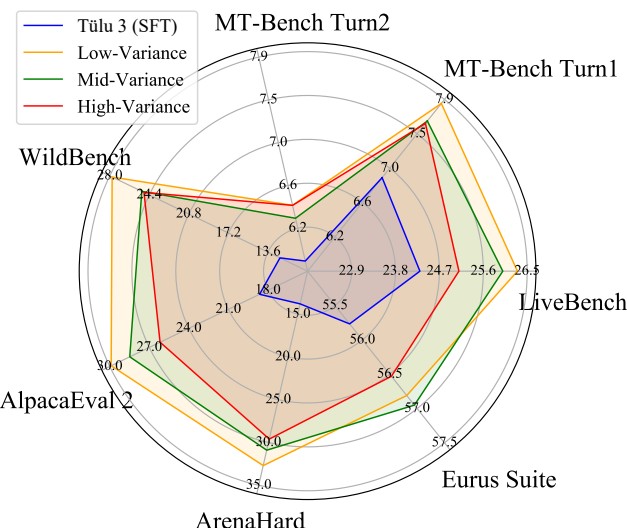

Figure 11: Performance across different benchmarks, comparing different response score variances. Qwen-2.5-72B-Instruct is employed for preference annotation while maintaining identical experimental setups to those specified in section 4.3.1.

