# OpenReview forum: "AIR: A Systematic Analysis of Annotations, Instructions, and Response Pairs in Preference Dataset"
_colmweb.org/COLM/2025/Conference — COLM 2025_

### Official Review · Reviewer_xkuT · 2025-05-13

**Rating:** 8
**Confidence:** 4
**Ethics Flag:** 1

**Summary:**

Preference learning for LLMs requires preference annotations, instructions, and response pairs; these three aspects are typically conflated in current work, which makes it difficult to disentangle their effects. The authors propose AIR, an analysis framework that attempts to isolate, analyze, and optimize for each of the three aspects separately. For Annotations, the authors investigate the effects of annotator model choice, prompt design, and scoring methods (e.g., pairwise vs scoring). For Instructions, they investigate score variance and number of prompt turns. For Response, they investigate how to construct pairs from scored examples, varying the score margins, thresholds, and on/off-policy mixing. Preference data is generated by 17 open-source LLMs; instructions through ShareGPT and UltraFeedback; all experiments are conducted on the Llama-3.1-Tulu-8B-SFT base model with 30k preference pairs and vanilla DPO; and alignment is assessed using 6 benchmark datasets across a range of domains. Based on experiments conducted using this framework, the authors recommend the following for alignment dataset design: (i) use point-wise scoring, (ii) select instructions with low score variance across different LLMs, and (iii) optimize for high quality response pairs with clear preference score margins. Combining all strategies yields substantial performance gains.

**Questions To Authors:**

- Some of the recommendations borne of this framework are dependent on each other (e.g., it makes sense to adjust score variance in the context of point-wise scoring, but this would not apply in settings whether the annotations are pairwise). It would be useful to provide some recommendations that generalize beyond the wholesale set of recommendations identified in the work, since there may be partial constraints on dataset collection. This could be in the context of future work (the manuscript is dense enough as is) and how the AIR framework allows for discovering other performant mixtures of strategies for data collection.
- In some cases, the impacts of combining different interacting strategies is not super clear, e.g., in Figure 4, how do relative score margins, score thresholds, and on/off-policy mixing strategies  interact? Are these plots made while holding other strategies constant?

**Reasons To Accept:**

- This manuscript contains a wealth of insights for preference dataset curation
- The framework is a useful way of disentangling different aspects of preference learning dataset curation. While not every combination of strategies is explored in this manuscript due to length, the framework will help others conduct such studies in the future.
- Experiments are thorough and support the findings; confirmatory experiments with another base model would be nice

**Reasons To Reject:**

N/A

---

> ### Author Response · Authors · 2025-06-01
> **Response by Authors**
>
> We sincerely thank you for your incredibly positive and constructive feedback! We are thrilled that you found our manuscript to contain "a wealth of insights for preference dataset curation" and recognized the AIR framework as a "useful way of disentangling different aspects of preference learning dataset curation". We will address your comments and questions in detail below:
>
> **Reasons To Accept #3**
> > Confirmatory experiments with another base model would be nice.
>
> Thank you for this important suggestion about the generalizability of our findings across different base models. This is indeed a common question raised by all reviewers, so please refer to our **Global Response for Using Another Base Model**.
>
> **Questions To Authors #1**
> > It would be useful to provide some recommendations that generalize beyond the wholesale set of recommendations identified in the work, since there may be partial constraints on dataset collection.
>
> We fully agree with the reviewer's insightful recommendation to provide guidance that generalizes beyond our 'wholesale set' of optimal recommendations, especially for scenarios with partial constraints on dataset collection. While our work identifies the most performant strategy when unconstrained, the fundamental strength of the AIR framework lies in its component-wise analysis. This systematic dissection allows practitioners to:
> - **Identify and work around constraints**: Even if certain components are fixed due to collection limitations (e.g., being restricted to pairwise annotations), the framework helps understand the expected performance impact of that constraint.
> - **Optimize unconstrained elements**: By understanding the isolated impact of each component, AIR enables targeted optimization of the remaining flexible elements based on our empirically validated principles.
> - **Discover new performant mixtures**: The framework inherently provides a methodology for systematically exploring and quantifying the trade-offs of different strategy combinations, thereby allowing for the discovery of other effective mixtures tailored to specific partial constraints.
>
> We will emphasize this practical utility more explicitly in the revised manuscript. This indeed opens up a crucial avenue for future work to formally explore and delineate these performant mixtures under various real-world constraints.
>
> **Questions To Authors #2**
> > In some cases, the impacts of combining different interacting strategies is not super clear, e.g., in Figure 4, how do relative score margins, score thresholds, and on/off-policy mixing strategies interact? Are these plots made while holding other strategies constant?
>
> We appreciate your providing this opportunity to clarify our experimental setup. We agree that the current description in the paper could be more explicit regarding the interactions and controlled variables.
>
> To confirm, all our experiments are conducted under a controlled variable methodology. For instance, when investigating the impact of absolute score threshold in response pair construction (as shown in Figure 4, middle plot), we define three settings: High, Mid, and Low, based on **the score of the chosen response** (line 292). For each chosen response, we then carefully select its corresponding rejected response. During this selection, we control the relative score margin to be between **Δ=2 or 3** for all pairs. Furthermore, for this specific analysis, all chosen and rejected responses are initially **off-policy**, meaning they are not generated by the model currently being trained.
>
> We acknowledge that this level of detail was not sufficiently clear in the original manuscript. In the revised version, we will enhance the clarity and detail of each experimental setup description to ensure that the interactions between strategies and the controlled variables are explicitly and comprehensively presented.

---

> > ### Author Response · Authors · 2025-06-03
> > **Response by Authors (Part 2)**
> >
> > **Reasons To Accept #2 & Questions To Authors #1 Follow-ups**
> > > Not every combination of strategies is explored in this manuscript due to length; Some of the recommendations borne of this framework are dependent on each other.
> >
> > Regarding your insightful point about the framework's utility in disentangling different aspects and the interdependencies of recommendations, we appreciate this feedback.
> >
> > In response to a similar query from Reviewer EMwy (**Reasons To Reject #1**), we have conducted additional preliminary interaction experiments during the rebuttal period. These experiments specifically explore the interplay between **Instruction Selection** and **Response Pair Construction** as a reference for interactions between dataset components. You can find the detailed results and analysis in our **Response by Authors (Part 1) to Reviewer EMwy**. We hope this additional analysis is helpful.

---

> > > ### Comment · Reviewer_xkuT · 2025-06-07
> > >
> > > I appreciate the additional experiments and response

---

> > > > ### Author Response · Authors · 2025-06-09
> > > >
> > > > Thank you for your positive feedback! We are delighted that the additional experiments and our response were helpful, and please do not hesitate to reach out if there are any remaining issues you would like to discuss.

---

### Official Review · Reviewer_E4jA · 2025-05-13

**Rating:** 6
**Confidence:** 4
**Ethics Flag:** 1

**Summary:**

This paper introduces the AIR (Annotations, Instructions, and Response Pairs) framework to systematically analyze the independent impacts of three key components—Annotations, Instructions, and Response Pairs—on aligning large language models (LLMs) with human preferences. Through experiments, it identifies three empirically validated principles: simplifying annotations with generative scoring, prioritizing low-variance instruction selection, and optimizing response pairs for quality and diversity. Aggregating all strategies, the authors observed substantial gain over baseline.

**Reasons To Accept:**

1. The authors tackle important problem. Constructing preference dataset has been a core component of language model alignment but hasn't been thoroughly investigated. This work investigate various strategies and execute a large number of ablation studies.

2. The resulting dataset outperforms baseline while only maintaining small portion.

**Reasons To Reject:**

The current paper presents a series of controlled experiments and their results. To enhance the robustness and credibility of the work, it would be beneficial to include explicit hypotheses for each experiment and additional supporting experiments to validate these hypotheses.

All experiments in the paper were conducted using a single model, specifically **Llama-3.1-Tulu-3-8B-SFT**. Expanding the scope to include more diverse models would significantly strengthen the findings. Models such as OLMo-2-1124-7B-SFT [1] or openbmb/MiniCPM-S-1B-sft [2] could serve as valuable candidates for additional experimentation.

Addressing these concerns would likely elevate the overall quality and impact of the paper, warranting a higher score, potentially up to 7.

[1] https://huggingface.co/allenai/OLMo-2-1124-7B-SFT

[2] https://huggingface.co/openbmb/MiniCPM-S-1B-sft

---

> ### Author Response · Authors · 2025-06-01
> **Response by Authors**
>
> We thank the reviewer for your thoughtful and constructive review! We are glad that you recognize our work for tackling an "important problem" and for thoroughly investigating various strategies through a "large number of ablation studies". We will address your comments and questions in detail below:
>
> **Reasons To Reject #1**
> > It would be beneficial to include explicit hypotheses for each experiment and additional supporting experiments to validate these hypotheses.
>
> We appreciate the reviewer's insightful suggestion regarding the inclusion of explicit hypotheses and additional supporting experiments to enhance the robustness and credibility of our work.
>
> We agree that a clear articulation of hypotheses is beneficial. In our current manuscript, our "Takeaways" (Section 4.2, 4.3, 4.4) function as empirically derived principles, which serve as our explicit hypotheses for each dataset component. Each of these "Takeaways" is investigated through a series of controlled experiments within Sections 4.2, 4.3, and 4.4, designed to validate its impact.
>
> Furthermore, to validate the synergistic effect of combining these principles, Section 4.5 ("Combined Impact") presents an overarching experiment demonstrating cumulative gains when integrating the optimal choices.
>
> Finally, our Cross-Verification (Section 4.6 and Appendix C.3) specifically aims to enhance robustness by validating our findings across different annotator models and instruction sources, thereby providing additional supporting evidence for the generalizability of our hypotheses.
>
> We would like to clarify if our understanding of "hypotheses" as these explicitly stated "Takeaways" is accurate. Please let us know if there is any misunderstanding on our part.
>
>
> **Reasons To Reject #2**
> > Expanding the scope to include more diverse models would significantly strengthen the findings.
>
> Thank you for this important question regarding the generalizability of our findings across different base models. This is indeed a common question raised by all reviewers, so please refer to our **Global Response for Using Another Base Model**.

---

> > ### Comment · Reviewer_E4jA · 2025-06-09
> >
> > Thank you for your detailed response and for conducting additional experiments during the rebuttal phase. I truly appreciate the efforts put forth by the authors.
> >
> > As mentioned, the additional experiments have demonstrated consistent results, which has prompted me to increase my score to 6.
> >
> > Regarding the term "hypothesis," I realize I may have misused it in my initial feedback. Throughout the paper, I notice a strong set of experimental results; however, the reasoning or underlying mechanisms behind these results are not clearly articulated.
> >
> > For instance, the result showing that Low variance filtering outperforms InsTag is intriguing, but the paper lacks an explanation or analysis as to why this might be the case. While the experiments and their results are indeed commendable, providing such reasoning or supporting claims would enhance the overall contribution and clarity of the work.

---

> > > ### Author Response · Authors · 2025-06-09
> > > **Response by Authors**
> > >
> > > Thank you for recognizing our efforts, pointing out this detail, and for the opportunity to explain our findings. We truly appreciate your kind words and the continued valuable feedback. We particularly acknowledge your insightful observation regarding the need for clearer articulation of the reasoning and underlying mechanisms behind our experimental results. We agree this is a crucial aspect for enhancing the overall contribution and clarity of our work.
> > >
> > > While our current work, the AIR framework, primarily adopts an empirical perspective to systematically explore and optimize preference dataset components, we have considered the underlying "why" for our key findings:
> > >
> > > - **Annotation Simplicity**: Our empirical results advocate for streamlined, greedy-decoded, point-wise generative scoring. This is because complex guidelines or multi-sample aggregation can inadvertently introduce noise or obscure the LLM annotator's holistic judgment, which is often more aligned with real-world preferences. A simpler signal provides cleaner input for the DPO objective, reducing distractions from non-essential annotation artifacts.
> > >
> > > - **Instruction Inference Stability**: Instructions with low variance in scores across different LLMs indicate that they reliably elicit fine-grained distinctions among responses. This consistent discriminability provides richer, more targeted signals for preference learning. In the DPO process, learning from such nuanced differences forces the model to refine its policy based on subtle quality gradients, which is more beneficial than learning from obvious errors (high variance) or indistinct prompts.
> > >
> > > - **Response Pair Quality**: Our findings suggest that optimal pairs (moderate margins, high absolute scores, hybrid mixing) collectively contribute to a robust contrastive learning signal for DPO. This is empirically observed as a balance: moderate margins provide clear yet diverse distinctions; high absolute scores ensure the learning is from competent responses; and hybrid mixing prevents overfitting to static distributions.
> > >
> > > These are qualitative possible reasons for how these choices shape the learning landscape for the model. We acknowledge that these explanations were not clearly articulated in the original paper. We will revise the manuscript to integrate these insights more explicitly, potentially in an expanded discussion section, to provide clearer reasoning for the observed phenomena. **However, these explanations are currently empirical and qualitative, not explained through quantitatively rigorous mathematical derivation.**
> > >
> > > **We focus on providing a concrete and robustly validated blueprint for constructing large-scale, high-quality automatically generated preference datasets, notably without reliance on closed-source models, thereby making a practical contribution to LLM preference learning.** We have considered that a deeper quantitative explanation of these phenomena might involve **analyzing the gradient towards preferred and dispreferred responses and monitoring the training dynamics in the DPO contrastive learning process**. We acknowledge that delving deeper into these theoretical mechanisms is a valuable and exciting direction for future work. This would provide a deeper, more theoretical understanding to complement our current empirical findings.
> > >
> > > Thank you once again for your constructive feedback, which continues to help us refine and strengthen our paper.

---

> > > > ### Comment · Reviewer_E4jA · 2025-06-09
> > > >
> > > > I appreciate the authors’ sincere effort in addressing my concerns.
> > > >
> > > > That said, I still feel that the explanations regarding the key findings tend more toward plausible conjectures rather than thorough, evidence-backed reasoning. For instance, the claim that optimal pairs collectively enhance contrastive learning remains somewhat speculative in its current form. If I were in the authors’ position, I might have considered illustrating this with visualizations in a simplified setting to provide more concrete intuition.
> > > >
> > > > To be clear, I believe the paper is ready for publication and makes a meaningful contribution. My intention is simply to highlight areas where the work could be further strengthened.

---

> > > > > ### Author Response · Authors · 2025-06-10
> > > > >
> > > > > Thank you for your very thoughtful response and for acknowledging our efforts. We are delighted that you find the paper "ready for publication" and making a "meaningful contribution." Your suggestion to illustrate concepts, such as how optimal pairs enhance contrastive learning with visualizations in a simplified setting, is incredibly insightful. We agree that this would provide more concrete intuition and significantly strengthen the work.
> > > > >
> > > > > We will take this valuable suggestion on board and explore it for the revised manuscript to enhance the clarity and depth of our explanations. Thank you for continuing to help us strengthen our paper.

---

> ### Author Response · Authors · 2025-06-07
> **Awaiting Your Feedback ~**
>
> As the discussion period is nearing its end, we would like to offer our continued availability to address any remaining questions or points of clarification you may have. Should there be any aspect we haven't fully addressed, or any misunderstandings on our part, please do not hesitate to let us know. We are committed to ensuring all your concerns are resolved.
>
> Thank you again for your valuable time and insightful feedback throughout this process.

---

> ### Comment · Area_Chair_uNtY · 2025-06-08
>
> Dear Reviewer E4jA, as the discussion period is drawing to a close, would you mind taking a look at the author response to see if it adequately addresses your concerns?
>
> Best,
> AC

---

### Official Review · Reviewer_EMwy · 2025-05-23

**Rating:** 7
**Confidence:** 4
**Ethics Flag:** 1

**Summary:**

This work systematically disentangle the impact of annotations, instructions and response pairs on the efficacy of preference datasets. They conduct controlled experiments to study each dataset component in open-source models. The work yields actionable insights to understand preference dataset design and can be useful for practitioners.

**Questions To Authors:**

1. Can you report the error bars across all results to better situate the performance differences with each component (Figure 2, Table 1would be most helpful) ?
2. In line 204, "Explanations/coarse guidelines offer negligible gains (≤ 0.28) over simpler baselines" why is this considered negligible gains when difference between Pair-guided and Pair-explained is 0.45 on average ? Did you implement a single-explained to better understand this effect ?
3. In line 207,  "The trend toward intricate annotation workflows may inadvertently introduce noise, advocating for streamlined designs." I am not convinced that this is a valid conclusion from Table 1. The difference in performance across various annotation designs are not well-investigated to understand whether this is noise or not.
4. Did you also study which RM (classifier or generative) had higher agreement with human annotations ?
5. How would these findings change or not change with the base model or a different architecture ?

These papers can be cited to better support this work:
1. Kim, Joongwon, et al. "A Systematic Examination of Preference Learning through the Lens of Instruction-Following." arXiv preprint arXiv:2412.15282 (2024).
2. Yin, Yueqin, et al. "Relative Preference Optimization: Enhancing LLM Alignment through Contrasting Responses across Identical and Diverse Prompts." arXiv preprint arXiv:2402.10958v1(2024).
3. Bansal, Hritik, John Dang, and Aditya Grover. "Peering through preferences: Unraveling feedback acquisition for aligning large language models." arXiv preprint arXiv:2308.15812 (2023).
4. Panickssery, Arjun, Samuel Bowman, and Shi Feng. "Llm evaluators recognize and favor their own generations." Advances in Neural Information Processing Systems 37 (2024): 68772-68802.

**Reasons To Accept:**

1. The controlled analysis of each component's impact allows for a systematic study and nuanced understanding of these components.
2. The work provides various actionable insights to improve performance such as low-variance instructions or generative-RM for scoring.
3. The experiments have good coverage of various components and design choices.
4. The takeaways are empirically validated across various benchmarks and cross-validates by replicating with a different instruction source and annotator model.

**Reasons To Reject:**

1. The study mainly focuses on the isolated impact of each dataset component and does not study the interactions between these components that may affect LLM performance.
2. While LLM based preferences are widely accepted, this work did not adequately study the impact of LLM annotations vs Human annotations and caveats with LLM annotations (for e.g. LLMs tend to favour their own generations) and how these impact preference dataset design.

---

> ### Author Response · Authors · 2025-06-03
> **Response by Authors (Part 1)**
>
> We sincerely thank the reviewer for your thorough and constructive feedback! We are very pleased that you recognize our work for its "systematic study and nuanced understanding" of preference dataset components. We will address your comments and questions in detail below:
>
> **Reasons To Reject # 1:**
> > The study does not study the interactions between these components that may affect LLM performance.
>
> We appreciate the reviewer's insightful comment regarding the limited study of interactions between dataset components. While our initial work focused on isolating individual impacts and demonstrated the cumulative effect of our findings in Section 4.5, we agree that a deeper understanding of component interactions is very important.
>
> We acknowledge that exhaustive pairwise interaction studies were not included in the original manuscript due to the substantial experimental cost (each setting requiring a DPO run and extensive evaluation, often with expensive APIs like GPT-4/GPT-4o). However, in direct response to your valuable feedback, we have conducted additional preliminary interaction experiments during the rebuttal period, specifically exploring the interplay between **Instruction Selection** and **Response Pair Construction**. Due to current experimental and evaluation API costs, these experiments were evaluated on **ArenaHard**, a benchmark widely recognized in recent work (e.g., Qwen 3).
>
> ***Exp 1: Instruction Variance Vs. Relative Score Margin***
>
> We divided instructions into low, mid, and high variance groups, consistent with Section 4.3.1. For each group, we investigated the optimal response pair score margin (based on margin distribution to balance pair count across groups for DPO training). We controlled other variables and evaluated results on ArenaHard.
>
> |                   | **Margin = 1** | **Margin = 2** | **Margin = 3/4** |
> | ----------------- | -------------- | -------------- | ---------------- |
> | **Low-Variance**  | 35.1           | 34.6           | 34.7             |
> | **Mid-Variance**  | 28.4           | 31.1           | 29.3             |
> | **High-Variance** | 29.3           | 31             | 31.2             |
>
> **Analysis**: We observe that while the optimal relative score margin can differ slightly for instructions with fixed variance, the superior performance of low-variance instructions remains consistently significant across all margin settings.
>
> ***Exp 2: Instruction Variance Vs. Absolute Score Threshold***
>
> Using the same instruction variance groups, we explored the optimal response absolute score threshold (low/mid/high settings consistent with Section 4.4.2, balancing pair count for DPO training). We controlled other variables and evaluated results on ArenaHard.
>
> |                   | **Low-Score** | **Mid-Score** | **High-Score** |
> | ----------------- | ------------- | ------------- | -------------- |
> | **Low-Variance**  | 34.3          | 34.7          | 35.2           |
> | **Mid-Variance**  | 27.7          | 27.9          | 31.7           |
> | **High-Variance** | 28.9          | 30            | 31.5           |
>
> **Analysis:** These results show that for instructions of any variance, selecting high-score responses consistently yields the best performance. Conversely, the benefit of low-variance instructions is consistently evident across all absolute score thresholds.
>
> These two experiments indeed demonstrate relevant interactions between instruction selection and response pair construction. However, they also reinforce our core conclusions: for instance, the consistent benefit of low-variance instructions, and that higher absolute scores are generally preferred. The specific optimal margin might vary, adding a layer of nuance. We will integrate these findings into the revised version of the paper, emphasizing these interactions and highlighting that a more exhaustive study of all component interactions remains a valuable direction for future work.

---

> ### Author Response · Authors · 2025-06-03
> **Response by Authors (Part 2)**
>
> **Reasons To Reject #2**
> > This work did not adequately study the impact of LLM annotations vs Human annotations and caveats with LLM annotations (for e.g. LLMs tend to favour their own generations) and how these impact preference dataset design.
>
> We appreciate the reviewer's insightful comment. We actually considered the potential for self-favoring bias and conducted a preliminary study. To investigate this, we sampled 1,000 instructions. For each instruction, responses were generated by both Meta-Llama-3.1-8B-Instruct and Qwen2-7B-Instruct. These responses were then judged independently by Meta-Llama-3.1-70B-Instruct and Qwen2-72B-Instruct. The win rates we observed were as follows:
> - Win rate (Llama as Annotator): {'Llama': 59.1%, 'Qwen': 40.9%}
> - Win rate (Qwen as Annotator): {'Llama': 55.1%, 'Qwen': 44.9%}
>
> While these results indeed suggest the presence of a self-bias effect, its impact was not overwhelmingly large. To further mitigate this, **we intentionally diversified our response generation pool beyond just the Llama and Qwen families**. As shown in Table 2, out of the 17 models we used for response generation, only 4 are Llama-based and 3 are Qwen-based. The remaining models, including the Gemma-2 series, Phi-3 series, Mistral series, MiniCPM3-4B, and DeepSeek-V2-Lite-Chat, belong to different architectural families. This broad diversity means any self-bias effect on our overall results is limited.
>
> Regarding human annotation, our work operates within the LLM-as-a-Judge paradigm due to the high cost and scalability challenges of human annotation for large datasets (e.g., ~30k preference pairs per DPO run). The paper's core contribution is optimizing these LLM-annotated dataset components. While direct comparison with human annotations and a comprehensive analysis of LLM judge biases are important, they were beyond the scope of this systematic analysis.
>
> **Questions To Authors #1**
> > Report the error bars.
>
> We appreciate the reviewer's excellent suggestion to include error bars. Firstly, it's important to distinguish between evaluation types. For **objective benchmarks** such as **Eurus** and **LiveBench**, where model responses are generated using **greedy decoding**, the evaluation results are deterministic and remain consistent across multiple runs. Therefore, variability (and thus error bars) is not applicable to these benchmarks.
>
> However, for **subjective LLM-as-a-Judge benchmarks** (MT-Bench, ArenaHard, AlpacaEval 2.0, WildBench (V2)), which utilize GPT-4o or GPT-4 series models for evaluation, results can exhibit some variability due to the nature of LLM judgment. As each single evaluation can incur significant costs (around $10), constrained by evaluation resources, we selected **ArenaHard** as a representative LLM-as-a-Judge benchmark to report error bars for the results presented in Figure 2 and Table 1.
>
> Here are the updated results for ArenaHard (totally **3 runs** including the original result, using a **95%** confidence level), reporting the average result and the confidence intervals:
>
> ***Figure 2 (Left)***
>
> |           | Llama-3.1-Tulu-3-8B-SFT | Generative RM    | Classifier-Based RM |
> | --------- | ----------------------- | ---------------- | ------------------- |
> | **ArenaHard** | 14.2 ($\pm$0.4526)         | 33.70 ($\pm$0.1960) | 32.13 ($\pm$1.1447)    |
>
> ***Figure 2 (Right)***
>
> |           | Llama-3.1-Tulu-3-8B-SFT | Single-Basic     | Single-Prob     | Single-Avg       |
> | --------- | ----------------------- | ---------------- | --------------- | ---------------- |
> | **ArenaHard** | 14.2 ($\pm$0.4526)         | 33.70 ($\pm$0.1960) | 34.1 ($\pm$0.6300) | 33.27 ($\pm$0.1307) |
>
> ***Table 1***
>
> |           | Llama-3.1-Tulu-3-8B-SFT | Single-Basic     | Pair-Basic       | Pair-Guided      | Pair-Explained   | Pair-Guided-Explained | Pair-Guided-Explained-FG |
> | --------- | ----------------------- | ---------------- | ---------------- | ---------------- | ---------------- | --------------------- | ------------------------ |
> | **ArenaHard** | 14.2 ($\pm$0.4526)         | 33.70 ($\pm$0.1960) | 32.06 ($\pm$0.7534) | 31.87 ($\pm$0.9076) | 31.50 ($\pm$1.1814) | 31.03 ($\pm$1.6016)      | 29.97 ($\pm$0.9216)         |
>
> These results demonstrate that the evaluations performed by GPT-4-0314 LLM-as-a-Judge are relatively stable, with confidence intervals indicating consistent performance across runs, even given the inherent variability of LLM-based judgments. The evaluations were conducted with a temperature of 0.6 and top_p of 0.95.

---

> ### Author Response · Authors · 2025-06-03
> **Response by Authors (Part 3)**
>
> **Questions To Authors #2**
> > In line 204, "Explanations/coarse guidelines offer negligible gains (≤ 0.28) over simpler baselines" why is this considered negligible gains? Did you implement a single-explained to better understand this effect?
>
> We thank the reviewer for pointing out this detail and for the opportunity to clarify our findings. Our statement that "Explanations/coarse guidelines offer negligible gains (≤0.28) over simpler baselines" (line 204) refers to the modest impact **when comparing Pair-Guided (41.30) and Pair-Explained (41.75) directly against Pair-Basic (41.47)**. Specifically, Pair-Guided (which introduces guidelines) showed a slight decrease of -0.17, while Pair-Explained (which requires an explanation first) showed a slight increase of +0.28. **This is distinct from the difference between Pair-Explained and Pair-Guided**.
>
> To further clarify this effect, we implemented and tested both Single-Explained and Single-Guided annotation strategies. The results, as **reported in Table 1 within the "Global Response for Using Another Base Model,"** show that Single-Guided (31.4354) and Single-Explained (31.2454) actually resulted in a decrease compared to Single-Basic (32.333).
>
> **Questions To Authors #3**
> > The difference in performance across various annotation designs are not well-investigated to understand whether this is noise or not.
>
> We thank the reviewer for prompting a deeper look into our conclusion regarding intricate annotation workflows introducing noise. To address your concerns directly, we have conducted further analyses for the revised manuscript:
> - We have now included **error bars** in some experimental results to quantify performance variability and consistency.
> - We also added experiments for **Single-Guided** and **Single-Explained** strategies to Table 1. These, combined with existing results, suggest that the observed patterns are less about random noise and more about the systematic behavior of LLM annotators.
> - Our existing **cross-verification (Section 4.6)** demonstrated consistent findings across different datasets and annotator models, which further supports the robustness of our conclusions.
> - We have also conducted a new experiment with an alternative base model, further reinforcing the generalizability of our principles. (As shown in **Global Response for Using Another Base Model**)
>
> Thank you for pushing us to strengthen this investigation.
>
> **Questions To Authors #4**
> > Did you also study which RM (classifier or generative) had higher agreement with human annotations?
>
> We appreciate this pertinent question. Our analysis in Section 4.2.1 focuses on which type of LLM-based RM yields better results when used to construct preference datasets for DPO training. However, this comparison was done on **downstream LLM alignment performance**, not framed as an "agreement with human annotations" study. We acknowledge that a direct investigation into the inter-annotator agreement between different LLM-based RMs and human judgments would provide valuable insights into the quality and biases of LLM annotations. This is an important area that we would consider for future work to further validate the reliability of LLM-as-a-judge paradigms.
>
> **Questions To Authors #5**
> > How would these findings change or not change with the base model or a different architecture?
>
> Thank you for this constructive suggestion regarding the generalizability of our findings across different base models. This is indeed a common question raised by all reviewers, so please refer to our **Global Response for Using Another Base Model**.
>
> **Questions To Authors #6**
> > These papers can be cited to better support this work.
>
> We thank the reviewer for these valuable citation suggestions. We agree they will strengthen our paper. This paper, Kim, Joongwon, et al. (2024), is already cited in line 104, relevant to our discussion on instruction selection. We will integrate the remaining three papers into our revised manuscript.

---

> ### Comment · Reviewer_EMwy · 2025-06-03
>
> Thank you for addressing my concerns and questions with a quick turnaround of experimental results. I also agree that the scope of the paper lies in LLM-as-a-Judge paradigm and thus, comparisons to human annotations are not needed in this work. I have also read the global response and convinced that this work's findings is generalizable to other model architectures. Overall I am satisfied with all the discussion and additional results, would love to see some of these incorporated in the final draft.

---

> > ### Author Response · Authors · 2025-06-03
> >
> > We sincerely thank the reviewer for your valuable feedback and for acknowledging our efforts in addressing the concerns and questions with a quick turnaround of experimental results. We are delighted that you are satisfied with the additional results, and we will ensure all discussed improvements are incorporated into the final draft.

---

### Author Response · Authors · 2025-06-01
**Global Response for Using Another Base Model (Part 1)**

We appreciate all reviewers' excellent questions regarding the generalizability of our findings across different base models or architectures:
- **Reviewer EMwy (Question #5)**: How would these findings change or not change with the base model or a different architecture?
- **Reviewer E4jA (Reasons To Reject #2)**: All experiments in the paper were conducted using a single model, specifically Llama-3.1-Tulu-3-8B-SFT. Expanding the scope to include more diverse models would significantly strengthen the findings.
- **Reviewer xkuT (Reasons To Accept #3)**: Confirmatory experiments with another base model would be nice.

This is indeed a critical aspect for establishing the robustness of our proposed principles. Firstly, our findings are already validated through cross-verification in Section 4.6, which demonstrated consistency when using a different annotator model and another instruction dataset.

To further strengthen the generalizability to diverse base models, and following Reviewer E4jA's suggestion, we conducted all isolated experiments using an additional base model: **OLMo-2-1124-7B-SFT.** Due to the significant experimental volume and evaluation API costs, we focused on this one alternative base model during the rebuttal period. We re-ran all isolated experiments presented in the main paper with OLMo-2-1124-7B-SFT to investigate the consistency of our conclusions. We are now reporting all these results.

**Figure 2 (Left):**
|                     | Average    | WildBench | MT-Bench  | Eurus Suite | LiveBench | ArenaHard |
| ------------------- | ---------- | --------- | --------- | ----------- | --------- | --------- |
| OLMo-2-1124-7B-SFT  | 28.1887    | 8.78      | 6.10625   | 43.701      | 17.9      | 9.5       |
| Generative RM       | **32.333** | **15.23** | **6.525** | **45.885**  | **19.6**  | **15.7**  |
| Classifier-Based RM | 30.9623    | 13.75     | 6.36875   | 44.474      | 18.4      | 14.5      |

**Figure 2 (Right):**
|                    | Average    | WildBench | MT-Bench  | Eurus Suite | LiveBench | ArenaHard |
| ------------------ | ---------- | --------- | --------- | ----------- | --------- | --------- |
| OLMo-2-1124-7B-SFT | 28.1887    | 8.78      | 6.10625   | 43.701      | 17.9      | 9.5       |
| Single-Avg         | 30.9154    | 13.25     | 6.325     | 44.277      | **19.8**  | 14        |
| Single-Prob        | 31.9186    | 14.02     | 6.475     | 45.223      | **19.8**  | **15.8**  |
| Single-Basic       | **32.333** | **15.23** | **6.525** | **45.885**  | 19.6      | 15.7      |

**Table 1 (Add Single-Guided & Single-Explained):**
|                          | Average    | WildBench | MT-Bench    | Eurus Suite | LiveBench | ArenaHard |
| ------------------------ | ---------- | --------- | ----------- | ----------- | --------- | --------- |
| OLMo-2-1124-7B-SFT       | 28.1887    | 8.78      | 6.10625     | 43.701      | 17.9      | 9.5       |
| Single-Basic             | **32.333** | **15.23** | 6.525       | **45.885**  | **19.6**  | **15.7**  |
| Single-Guided            | 31.4354    | 14.53     | 6.4625      | 45.822      | 18.3      | 13.9      |
| Single-Explained         | 31.2454    | 15.07     | 6.2625      | 45.132      | 18.5      | 14.9      |
| Pair-Basic               | 31.2866    | 12.7      | 6.4         | 44.833      | 19.5      | 15.4      |
| Pair-Guided              | 31.9255    | 14.63     | **6.53125** | 45.185      | 19.1      | 15.4      |
| Pair-Explained           | 31.4544    | 13.97     | 6.5         | 45.002      | 17.8      | 15.5      |
| Pair-Guided-Explained    | 30.8941    | 12.63     | 6.41875     | 44.453      | 18.8      | 14.4      |
| Pair-Guided-Explained-FG | 30.5774    | 13.38     | 6.1375      | 44.532      | 18.9      | 14.7      |

---

> ### Author Response · Authors · 2025-06-01
> **Global Response for Using Another Base Model (Part 2)**
>
> **Figure 3 (Left):**
> |                          | Average     | WildBench | MT-Bench    | Eurus Suite | LiveBench | ArenaHard |
> | ------------------------ | ----------- | --------- | ----------- | ----------- | --------- | --------- |
> | OLMo-2-1124-7B-SFT       | 28.1887     | 8.78      | 6.10625     | 43.701      | 17.9      | 9.5       |
> | Low-Variance             | **31.8957** | 13.39     | **6.56875** | **45.101**  | **19.9**  | **15.4**  |
> | Mid-Variance             | 30.7187     | 12.65     | 6.49375     | 43.706      | 18.9      | 13.4      |
> | High-Variance            | 30.4576     | 12.42     | 6.5125      | 43.943      | 18.7      | 12.1      |
> | InsTag Quality Filtering | 31.3266     | **13.41** | 6.55        | 44.323      | 18.7      | 14.7      |
>
> **Figure 3 (Right):**
> |                    | Average | WildBench | MT-Bench Turn 1 | MT-Bench Turn 2 | Eurus Suite | LiveBench | ArenaHard |
> | ------------------ | ------- | --------- | --------------- | --------------- | ----------- | --------- | --------- |
> | OLMo-2-1124-7B-SFT | 28.1887 | 8.78      | 6.625           | 5.5875          | 43.701      | 17.9      | 9.5       |
> | Single-Turn        | 30.3146 | 13.67     | 6.375           | 5.675           | 44.653      | 20.3      | 12.7      |
> | Multi-Turn         | 30.2908 | 12.1      | 6.625           | **5.925**       | 44.204      | 19.5      | 12.9      |
>
> **Figure 4 (Left):**
> |                    | Average    | WildBench | MT-Bench  | Eurus Suite | LiveBench | ArenaHard |
> | ------------------ | ---------- | --------- | --------- | ----------- | --------- | --------- |
> | OLMo-2-1124-7B-SFT | 28.1887    | 8.78      | 6.10625   | 43.701      | 17.9      | 9.5       |
> | Low Margin         | 30.4119    | 13.27     | 6.18125   | 43.577      | 18.6      | 14.8      |
> | Moderate Margin    | **31.689** | **14.22** | **6.425** | **46.275**  | 18.7      | **15**    |
> | High Margin        | 30.3838    | 12.76     | 6.2375    | 45.084      | **18.8**  | 12.9      |
>
> **Figure 4 (Middle):**
> |                     | Average     | WildBench | MT-Bench    | Eurus Suite | LiveBench | ArenaHard |
> | ------------------- | ----------- | --------- | ----------- | ----------- | --------- | --------- |
> | OLMo-2-1124-7B-SFT  | 28.1887     | 8.78      | 6.10625     | 43.701      | 17.9      | 9.5       |
> | High-Score Response | **31.4451** | **14.03** | **6.38125** | **44.283**  | **19**    | **16.1**  |
> | Mid-Score Response  | 30.4395     | 13.02     | 6.18125     | 43.865      | 18.4      | 15.1      |
> | Low-Score Response  | 29.5406     | 11.08     | 6.1875      | 44.048      | 18.4      | 12.3      |
>
> **Figure 4 (Right):**
> |                    | Average     | WildBench | MT-Bench   | Eurus Suite | LiveBench | ArenaHard |
> | ------------------ | ----------- | --------- | ---------- | ----------- | --------- | --------- |
> | OLMo-2-1124-7B-SFT | 28.1887     | 8.78      | 6.10625    | 43.701      | 17.9      | 9.5       |
> | Mid-Mix            | **32.3839** | **15.96** | 6.36875    | **45.072**  | **19.6**  | **17.6**  |
> | Low-Mix            | 31.7092     | 15.24     | **6.4125** | 44.881      | 19.1      | 15.2      |
> | Pure-Off-Policy    | 30.8596     | 13.72     | 6.225      | 44.628      | 18.6      | 15.1      |
> | Pure-On-Policy     | 31.0277     | 14.8      | 6.09375    | 44.401      | 18.4      | 16.6      |
>
> As can be seen, the experimental results from Sections 4.2, 4.3, and 4.4 in the main text show significant consistency for OLMo-2-1124-7B-SFT, with the average results being consistent. This indicates that our core findings generalize well across different base model architectures.
>
> We will integrate these new base model experimental results into the revised version of our paper to further highlight the credibility of our findings.

---

### Decision · Program_Chairs · 2025-07-08

**Decision:**

Accept

**Comment:**

The reviewers all agreed that this is a solid study of what makes for a good preference tuning dataset. The takeaways from the careful experiments done in this paper help to crystallize (and/or refute) common wisdom in the field, and they will be very helpful for researchers and developers building preference datasets. I encourage the authors to update their manuscript to account for the reviewer comments, e.g., the addition of the new base model and clarifications on the scope.